# A quantitative analysis of the impact of explicit incorporation of recency, seasonality and model specialization into fine-grained tourism demand prediction models

**Amir Khatibi**⊙*, **Ana Paula Couto da Silva**⊙, **Jussara M. Almeida**⊙, **Marcos A. Gonçalves**⊙

Department of Computer Science, Federal University of Minas Gerais (UFMG), Belo Horizonte, Brazil

⊙ These authors contributed equally to this work.
* amirkm@dcc.ufmg.br

**Data Availability Statement:** All the data exploited in our paper is already published and publicly

## Abstract

Forecasting is of utmost importance for the Tourism Industry. The development of models to predict visitation demand to specific places is essential to formulate adequate tourism development plans and policies. Yet, only a handful of models deal with the hard problem of fine-grained (per attraction) tourism demand prediction. In this paper, we argue that three key requirements of this type of application should be fulfilled: (i) recency—forecasting models should consider the impact of recent events (e.g. weather change, epidemics and pandemics); (ii) seasonality—tourism behavior is inherently seasonal; and (iii) model specialization—individual attractions may have very specific idiosyncratic patterns of visitations that should be taken into account. These three key requirements should be considered *explicitly and in conjunction* to advance the state-of-the-art in tourism prediction models. In our experiments, considering a rich set of indoor and outdoor attractions with environmental and social data, the explicit incorporation of such requirements as features into the models improved the rate of highly accurate predictions by more than 320% when compared to the current state-of-the-art in the field. Moreover, they also help to solve very difficult prediction cases, previously poorly solved by the current models. We also investigate the performance of the models in the (simulated) scenarios in which it is impossible to fulfill all three requirements—for instance, when there is not enough historical data for an attraction to capture seasonality. All in all, the main contributions of this paper are the proposal and evaluation of a new information architecture for fine-grained tourism demand prediction models as well as a quantification of the impact of each of the three aforementioned factors on the accuracy of the learned models. Our results have both theoretical and practical implications towards solving important touristic business demands.

available in Mendeley Repository: https://data.
mendeley.com/datasets/t7bfhtzhxg/1.

**Funding:** This research is partially funded by
Conselho Nacional de Desenvolvimento Científico e
Tecnológico (CNPQ) and Fundação de Amparo à
Pesquisa do Estado de Minas Gerais (FAPEMIG).
The authors and grant numbers are as following: ●
Amir Khatibi: (CNPq: 169823/2017-2) ● Ana Paula
Couto da Silva and Marcos A. Gonçalves:
(FAPEMIG: PPM-00543-17 and PPM-00177-18),
(CNPq: 422593/2018-4, 310538/2020-3, 310668/
2020-4, 402711/2021-1 and 403184/2021-5) ●
Jussara Almeida: (CNPq: 305683/2019-5 and
403106/2021-4) The funders had no role in study
design, data collection and analysis, decision to
publish, or preparation of the manuscript.

**Competing interests:** The authors have declared
that no competing interests exist.

# 1 Introduction

According to the World Travel and Tourism Council (WTTC), as of 2019 annual research covering 185 countries and economies, the global travel and tourism contribution to the Gross Domestic Product (GDP: a monetary measure of the market value of all the final goods and services produced in a specific time period) is at 10.3% supporting 319 million jobs. This corresponds to 10% of the global employment. Considering new jobs across the world, the contribution of travel and tourism industry is even higher, achieving 25% of all global new jobs created over the last five years (reported by Global Economic Impact and Trends 2020 accessible at link https://wttc.org/Research/Economic-Impact). Thus having estimated values of future tourism demand in the weeks, months, and years ahead can serve as a base for preparing activities necessary for creating comprehensive tourism policies [1].

The importance of accurate tourism prediction becomes indisputable when ones realizes that tourism products are generally perishable—unsold flight seats, empty hotel rooms and unsold tickets of a tourism attraction are just a few examples. In addition, tourism demands are sensible to factors like exchange rate [2], fuel price, climate changes [3], local and global financial crises [4] and even epidemics/ pandemics. The new Coronavirus Disease (COVID-19) (more information about this disease at link https://www.who.int/emergencies/diseases/novel-coronavirus-2019/technical-guidance) pandemic has completely shut down the Tourism Industry worldwide. Accurate forecasting could have helped to deal with the crisis letting a better management of the initial sector's recovery. There has been a real need to develop robust prediction models that not only forecast well the future visits by considering seasonal aspects of tourism behaviour but also show flexibility to recent trends and events and idiosyncratic aspects of the attractions.

In our previous work [5], the effects of environmental and social media data over tourism visitation have been studied in two scenarios—indoor and outdoor attractions. Visitation census, environmental features and social media data, for 27 museums and galleries in the United Kingdom (indoor attractions) as well as 76 national parks in the United States (outdoor attractions) have been exploited. Our proposal showed superior prediction accuracy when compared to the State-Of-The-Art (SOTA) results using features from both social media and environmental data adopting various prediction models and exploiting different modeling approaches.

In our previous analysis, it was observed that for outdoor attractions, environmental features have better predictive power while the social media features have more influence in the case of indoor attractions. In any case, best results, in all scenarios, were obtained when using both types of features jointly as input to a Support Vector Regression (SVR) prediction model obtaining moderate or highly accurate prediction results for around 93% of the attractions.

In this work, a new tourism prediction methodology is proposed that **explicitly** incorporates aspects related to recency, seasonality and specialization into the prediction models. More than explicitly considering such requirements into our information architecture—something that previous work has not done—in our current study we **quantify the impact** of each one of these effects as well as their interactions for fine-grained high-accuracy tourism demand prediction task while also improving our previous (state-of-the-art) results in tourism prediction [5]. We do this by arguing and demonstrating that three other key requirements of (1) *recency*, (2) *seasonality* and (3) *model specialization* should be fulfilled by an accurate model. These requirements should be captured as *explicit features* or properties of models for tourism forecasting, something that the previous state-of-the-art has not explored. Though these characteristics have been considered to different extents in different solutions in isolation, we are the first to consider them altogether as essential aspects that should be explicitly Incorporated

in conjunction into prediction model for fine-grained (attraction-level) tourism demand prediction. Our work is the first to measure and quantify the isolated and combined impact of these factors when incorporated into the models. These requirements are briefly introduced next.

**Recency** considers the impact of recent events on the prediction models. Prior work mostly focuses on the importance of seasonality as the main temporal aspect for tourism prediction. We argue that other temporal aspects should be considered to assess whether and how recent events such as financial crises, new trends, epidemics/ pandemics, new infrastructures, may impact predictions.

A few prior studies analyze the effect of recency on tourism demands. In [6] and [7], the authors study temporal aspects considered as important to predict tourism visits. Particularly, in [6], an algorithm for recommending personalized tours is proposed using users' recent preferences as one of the variables of their model. In their tour recommendation algorithm, they enhance the models by a weighted update of user interests based on the recency of their visits giving more emphasis to more recent Point of Interest (PoI: an entity of interest with well-defined location for example museums, churches, waterfalls and coffee shops) visits. They show improvements upon earlier tour recommendation work.

Though prediction models such as Auto Regressive Integrated Moving Average—ARIMA-based [8] ones indirectly exploit recency by means of temporal series modeling, our argument is that recency should be promoted as an *explicit first-class feature* to be incorporated into the prediction models. First-class features are input features that in other models such as ARIMA-based ones are captured implicitly.

**Seasonality** focuses on the inherently cyclic behaviour of tourism demands. Several studies in the literature focus on the importance of seasonality as the main temporal aspect for tourism prediction. Seasonality has been defined as the inherently cyclic behaviour of tourism demands. The authors of [9] state that seasonality is one of the main phenomena affecting tourism. According to them, seasonality is the systematic, although not necessarily regular, intra-year movement caused by changes in the weather, the calendar, and timing of decisions made by the agents of the economy, directly or indirectly through the production and consumption decisions. The authors of [10], instead, explain seasonality as a temporal imbalance in the phenomenon of tourism, which may be expressed in terms of dimensions of such elements as numbers of visitors, expenditure of visitors, traffic on highways and other forms of transportation, employment, and admissions to attractions.

In [11], the authors state that, regarding periodicity, the main focus of interest had been annual seasonality, with studies that show the differences in tourism activity between different seasons. In contrast, in their work, they perform a decomposition analysis of yearly, monthly and weekly seasonalities of tourism demand. They do so by conducting an in-depth analysis of intra-monthly and intra-weekly tourism demands using entropy and relative redundancy measures. The authors show that seasonality is present in annual, monthly and weekly frequencies using the Balearic Islands airports as their case study. In addition, they show that monthly and weekly seasonality differs across geographical markets. Since variations during the year are often caused by the climate or other social factors, intra-monthly and intra-weekly changes in tourism demand should be more closely associated with institutional or social factors, due to non-working days during the week, work holidays and other events that take place at specific times, such as Christmas, school or university holidays and work vacations.

The authors of [12] focus their study on the impact of seasonality on cultural tourism—defined as tourism focused on cultural motivations, including visits to museums and archaeological sites. They analyze tourism seasonality in some selected destinations in Sicily, concluding that cultural destinations are less impacted by seasonality in tourism flows.

In a recent survey on tourism forecasting [7], the authors state that the most widely adopted statistical time series prediction method is the seasonal auto-regressive integrated moving average—SARIMA. They claim that SARIMA is able to capture seasonality and recency implicitly in their forecasts. SARIMA is one of our baselines serving the purpose of comparing explicit versus implicit modeling of such requirements.

Our experiments aim to demonstrate that explicitly exploiting seasonality can greatly improve the prediction accuracy.

**Model specialization** advocates creating specialized individual models for each touristic attraction. The main motivation is that particular attractions may have very specific intrinsic patterns of visitations. Studies such as [13, 14] explore the tourists' motivations in the process of attraction selection. For instance, the authors of [13] identify different motivations and behavioral patterns in visits to different types of museums. For example, tourists visiting the historic Rembrandt House were more likely to be accompanying other people, more likely to want to learn new things, as well as more likely to be in search of entertainment or local culture and history than those interviewed at the Stedelijk museum of modern art. They also find distance of the visitor from the origin, geographical origin of tourists, their socio-demographic characteristics, travel form and the period of staying in the destination are also important factors affecting the choice of attractions to visit. On the other hand, in [14], the authors study the generation Y preferences (generation Y is the generation born in the 1980s and 1990s, comprising primarily the children of the baby boomers and typically perceived as increasingly familiar with digital and electronic technology), finding that this generation has his own profile and patterns of consumption. They discuss money spending preferences, the technology facilities in the attractions, the design of the place and the presence of information in social media as some of motivational differences. All in all, this serves as another factor that can affect differently the visitation patterns of different attractions motivating specialized models of visitation for each attraction. However, creating specialized individual model for each attraction can be advantageous since individual attractions may have very specific idiosyncratic patterns of visitations. On the other hand, there may be cases when one may not have enough data to train an individual model for each site. In this case, it is more viable to train single models for attractions of a given type to benefit from a vast amount of available social, climate and official data in the training process. Our experiments analyze this trade-off in depth.

In our work, we aim to demonstrate that, by explicitly exploiting the three proposed key requirements of tourism prediction as features, our models can greatly improve prediction accuracy regarding the SOTA results. Indeed, our experimental results, considering a rich set of indoor and outdoor attractions with environmental and social data, show that the explicit incorporation of such requirements into the models can improve the rate of *highly accurate predictions* by more than 320% against the current SOTA [5]. Our proposed models can even help to solve difficult prediction cases, poorly solved by the current solutions. For instance, the National Portrait Gallery in the U.K. saw a huge increase in social media reviews (over 50% by April 2015) but that was not accompanied by real world visits, causing the models to mistakenly follow the social patterns, ultimately implying in low accuracy. Another example is the Bryce Canyon national park in the U.S., in which the visits experience some period of untypical increases (more than 20% in Feb. to Sep. 2016 in comparison with the same period in 2015). That increase was not reflected neither in the environmental features nor in the social media reviews, both inputs exploited by the SOTA model. These situations can be dealt with by explicitly incorporating recency and seasonality features.

## 1.1 Research questions

The main hypotheses of our work, posed as explicit research question to be answered, include:

***RQ 1: Do recency, seasonality and model specialization (characteristic of attraction) influence the accuracy of predicting visits in tourist sites?*** Adopting our collected rich set of indoor and outdoor attractions, our specialized and global prediction models explicitly exploit recency and seasonality features in order to evaluate whether each of the key requirements of tourism prediction influence the accuracy of the models.

***RQ 2: What is the impact of each of recency, seasonality and model specialization in tourism demand prediction?*** In order to quantify the isolated and combined impact of each of these three requirements (recency, seasonality, and specialization), a factorial design analysis [15] is applied. In this analysis, the impact of each of the three requirements is evaluated in two different scenarios of attractions: outdoors (parks) and indoors (museums). In both scenarios seasonality, model specialization and the interaction between them have the largest impact in the prediction accuracy, with seasonality being more important in the case of outdoors. It was observed that model specialization is the most prominent factor to improve results, mainly for highly accurate predictions.

***RQ 3: How scenarios with data scarcity hinder the accuracy of prediction models while exploiting recency, seasonality and model specialization?*** Our results show that the absence of recency or seasonality features drastically reduces the accuracy of prediction models in scenarios with data scarcity. Recency features are not as important as the seasonality ones, but they still have a relevant impact on prediction accuracy, mainly for situations in which there is not enough historical data to capture seasonality for a given attraction.

## 1.2 Related work

Table 1 summarizes the main related studies that exploit in one way or another the aforementioned tourism key requirements. The table highlights, for each work; (i) whether the work applies the proposed techniques in multiple attractions or are concentrated in only one specific case study such as a country or a single touristic site; (ii) whether the work uses external features (data) as a proxy to predict the visitations, for instance, the use of socio-economic or environmental features, and (iii) whether they explicitly explore recency and/or seasonality.

## 1.3 Contributions and outline of the paper

To summarize, the main contributions of this article are:

- **Section 2** presents our collected dataset specification, followed by problem definition, short description of exploited prediction techniques, experimental methodology, features exploited in the prediction models, and the evaluation metrics. The investigation of all proposed **RQs** requires a rich dataset to permit in-depth analysis of the effects of tourism requirements in multiple category of attractions.

- **Section 3** aims to experimentally answer our posed Research Questions (RQs) for tourism attractions. Sections 3.1 and 3.2 investigate **RQ1** demonstrating that the three tourism key requirements, i.e. recency, seasonality and model specialization are essential for fine-grained high-accuracy tourism demand prediction task. More than that, these requirements should be incorporated as explicit features into the learning models. Our experimental evaluation confirms our hypotheses, with observed gains over the other solutions. We also show

**Table 1. Related work and our contributions.**

| Related Work | Work Domain | Multiple Attractions | External Data | Explicit Recency | Explicit Seasonality | Our work |
|---|---|---|---|---|---|---|
| [16] | Predicting coarse-grained tourism demand for entire country Turkey using multiple socio-economic features | ✗ | ✓ | ✗ | ✗ | Use of environmental and social media features in a fine-grained prediction level for multiple attractions |
| [17] | Use of Wikipedia usage trends in order to forecast tourism demand of Hawaii reporting the accuracy of their prediction results only by Root Mean Squared Error (RMSE) | ✗ | ✓ | ✗ | ✗ | Use of environmental and social media features in more than 100 outdoor and indoor attractions |
| [18] | Analyses the relationship between the internet search data (Baidu in China) and the actual tourist flow only for a single city, Beijing Forbidden City | ✗ | ✓ | ✗ | ✗ | Extended study of tens of attractions divided into two groups, studying the performance of different classes of features |
| [19] | Use of Location Based Social Networks (LBSN) to study mobility of tourists and citizens in a coarse-grained fashion | ✗ | ✓ | ✗ | ✗ | Fine-grained analysis in social media networks |
| [20] | Uses the locations of photographs in Flickr to estimate visitation counts in some recreational sites | ✓ | ✓ | ✗ | ✗ | Improving the accuracy of prediction models exploiting environmental features alongside explicit use of recency and seasonality factors |
| [21] | Analyses the climate and visitation data for the U.S. national parks using a single model of third-order polynomial temperature model with an accuracy of 69% | ✓ | ✓ | ✗ | ✗ | Use of multiple prediction models exploiting social media features alongside explicit use of recency and seasonality factors |
| [22] | Exploits travellers' Google web search and history of tourism arrivals to analyze temporal relationships between search terms and tourist arrivals in a single attraction (a Swedish mountain) | ✓ | ✓ | ✗ | ✗ | Quantification of performance of the explicit use of recency and seasonality factors for more than 100 attractions while improving the accuracy of results by adding environmental features and other types of external data |
| [23] | Use of search engine data with a de-noising step in order to avoid misleading or invalid predictions by comparing the performance of different noise-processing techniques only in Jiuzhaigou park in China | ✓ | ✓ | ✗ | ✗ | Analysis of more than 100 attractions with different characteristics in two categories of indoors and outdoors in a fine-grained manner |
| [24] | Tourism demand Prediction of top five most visited museums in London with free admission evaluating different algorithms exploiting the Google Trends index as the main feature | ✓ | ✓ | ✗ | ✗ | Compared to our work, the former is very limited in terms of the type of attraction, location and exploited features. while their main feature Google Trends index is a black-box with proved probability of overestimation problem |
| [9] | Analyse seasonality as one of the main phenomena affecting tourism, i.e systematic, although not necessarily regular, intra-year movement caused by changes in the weather, the calendar, and timing of decisions made by the agents of the economy | ✗ | ✗ | ✗ | ✓ | In contrast to this work, our contribution is in explicit use of recency factor besides exploiting external data to improve the accuracy of fine-grained prediction models |
| [10] | Analyse seasonality as a temporal imbalance in the phenomenon of tourism, which may be expressed in terms of dimensions of such elements as numbers of visitors, expenditure of visitors, traffic on highways, employment, and admissions to attractions | ✗ | ✗ | ✗ | ✓ | Fine-grained analysis and quantification of the effects of the recency factor, isolatedly and conjointly with seasonal factors in two different class of indoor and outdoor attractions |
| [11] | Perform a decomposition analysis of yearly, monthly and weekly seasonalities of tourism demand showing that seasonality is present in annual, monthly and weekly frequencies using the Balearic Islands airports as their single case study | ✓ | ✗ | ✗ | ✓ | A complete factorial design analysis for quantifying the effect of the recency factor in specialized and global models separately for two types of attractions |

*(Continued)*

**Table 1.** (Continued)

| Related Work | Work Domain | Multiple Attractions | External Data | Explicit Recency | Explicit Seasonality | Our work |
|---|---|---|---|---|---|---|
| [12] | Study the impact of seasonality on cultural tourism—defined as tourism focused on cultural motivations, including visits to museums and archaeological sites. They analyze tourism seasonality in some selected destinations in Sicily, concluding that cultural destinations are less impacted by seasonality in tourism flows | ✓ | ✗ | ✗ | ✓ | our work not only quantifies the effects of recency and seasonality factors in isolation and conjointly but also presents improvements in performance for our predictions exploiting external features regarding social media and environmental data using dozens of attractions |
| [7] | Analyse and state that the most widely adopted statistical time series prediction method is the SARIMA which is able to capture seasonality and recency implicitly in forecasts | ✗ | ✗ | ✗ | ✗ | Analysis of multiple prediction models exploiting external features of social media and environmental alongside the explicit use of recency and seasonality in specialized models. SARIMA is used as one of our baselines. |
| [6] | Propose an algorithm for recommending personalized tours based on users' recent preferences as one of the variables of their model enhancing their models by a weighted update of user interests based on the recency of their visits giving more emphasis to more recent visits | ✗ | ✗ | ✓ | ✗ | Our focus is in the task of tourism demand prediction and not on recommendation of touristic sites, though recency and seasonality features are explored as in that work. Our work also quantifies the effect of each of the factors for two classes of indoor and outdoor attractions |
| [25] | Investigate experiences of Chinese economy hotel guests using online reviews as proxy. Applies a deep learning fine-grained sentiment analysis to rank each of positive and negative sentiments associated with tourists sentiments such as location, facilities, service, price, image, sound insulation. | ✓ | ✓ | ✗ | ✗ | Similarly to our work, the authors use external data for fine-grained predictions. However, the focus of our work is on prediction of visits instead of sentiment analysis. Our work also exploits recent and seasonal behaviors explicitly in our feature-set. |
| [26] | Develops a scalable online platform for extracting, analyzing, and sharing multi-source multi-scale human mobility flows to assist human mobility monitoring and analysis during disaster events such as the ongoing COVID-19 pandemic in understanding human mobility dynamics. | ✓ | ✓ | ✗ | ✗ | The focus of this work is mostly on providing and monitoring fine-grained spatio-temporal mobility data while our work analyses multiple prediction models exploiting external data alongside explicit use of seasonality and recency in order to predict tourism demand. |
| [27] | Builds a fine-grained tourist satisfaction prediction model based on deep learning, using features such as "location, service, cost performance, environment, facilities and others" of the destination, and their division into several fine-grained dimensions. | ✓ | ✓ | ✗ | ✗ | Similarly to our work, the authors use external data for fine-grained prediction. However the focus of our work is in visits prediction instead of tourist satisfaction prediction. In addition, Our work exploits recent and seasonal features to obtain more accurate results. |

that the explicit incorporation of such requirements into the models help to solve very hard-to-solve cases.

Regarding **RQ2**, a factorial analysis is performed in order to quantify the impact of each of the three requirements on the accuracy of the learned models in indoor and outdoor attractions. The analyses show that the most impacting ones are model specialization and seasonality but recency is effective when there is not enough historical data about a specific attraction.

To investigate the **RQ3**, a study on the performance of each of the tourism prediction requirements is performed for cases with no recent or historical data for a given attraction. The study shows that the absence of recency or seasonality features drastically reduces the accuracy of prediction models in different scenarios.

- **Section 4** discusses and analyses our achieved experimental results, connecting them with the posed Research Questions. It provides insights regarding why our proposed methods work the way they did based on our interpretations of all performed analyses.

- **Section 5** summarizes our results to answer RQs. It emphasizes that the best results are obtained when using all types of features, i.e. external data features jointly with key tourism requirements. Conclusions and directions for future work are also provided.

All in all, the main contributions of this paper are (i) a new information architecture that **explicitly incorporates three new factors** into the state-of-the-art fine-grained tourism prediction models, greatly improving model accuracy and (ii) a **quantification** of the impact of each of the three factors on the accuracy of the learned models. *Our results have both theoretical and practical implications towards solving important touristic business demands.*

## 2 Materials and methods

In this section, we elaborate our experimental methodology for the task of evaluating the role of recency, seasonality and model specialization in tourism demand prediction. We first review the adopted datasets and then present the problem formulation. Next, the exploited prediction techniques are explained, offering a brief description of each of them. The learning and parameterization of the prediction models are discussed next. Finally, the prediction architecture is illustrated along with the definition of the factors for the factorial design analysis.

### 2.1 Datasets

Our present work relies on our previously published FISETIO dataset [28] for experimental analysis. This dataset was collected from five official and governmental sources: (1) the U.S. National Park Service was selected as the main source for the official data for the outdoor tourism demands; (2) TripAdvisor was used as the source for the social media related features of the outdoor and indoor dataset; (3) U.S. national climate data center was used as the origin of the climate data for the outdoor attractions; (4) the Department for Digital, Culture, Media and Sport of England providing the official visits for the indoor attractions and (5) the U.K. national weather service (Met Office) to gather weather conditions for indoor dataset.

We collected, cleaned and merged all data into two categories of attractions, namely, outdoors and indoors. Fig 1 illustrates the data collection phases to obtain the indoor and outdoor datasets for our analysis. Table 2 provides the sources and features -social media and environmental- in our datasets in brief. The reader is referred to our published dataset paper [28] for a detailed description of data sources and the data collection, data cleaning and the data integration processes. The data collection method is in compliance with the terms and conditions of the data provider in this case Mendeley Repository.

### 2.2 Problem Definition

The faced problem is forecasting the visitation for fine-grained touristic points. Yet, in addition to the social media and environmental features exploited in our previous work [5], we here also incorporate both recency and seasonality requirements. Moreover, we consider per-attraction model specialization. Given a touristic attraction, the prediction problem is defined as follows.

First, equally spaced non-overlapping time windows are created with the same temporal granularity (e.g., a month, a week, a day, an hour, etc) for each time-series of the variables in social media, environmental data, recency and seasonality features in the format of $X = \{X_1, X_2, \ldots, X_m\}$ where $m$ is the number of features. These time-series (e.g., number of reviews,

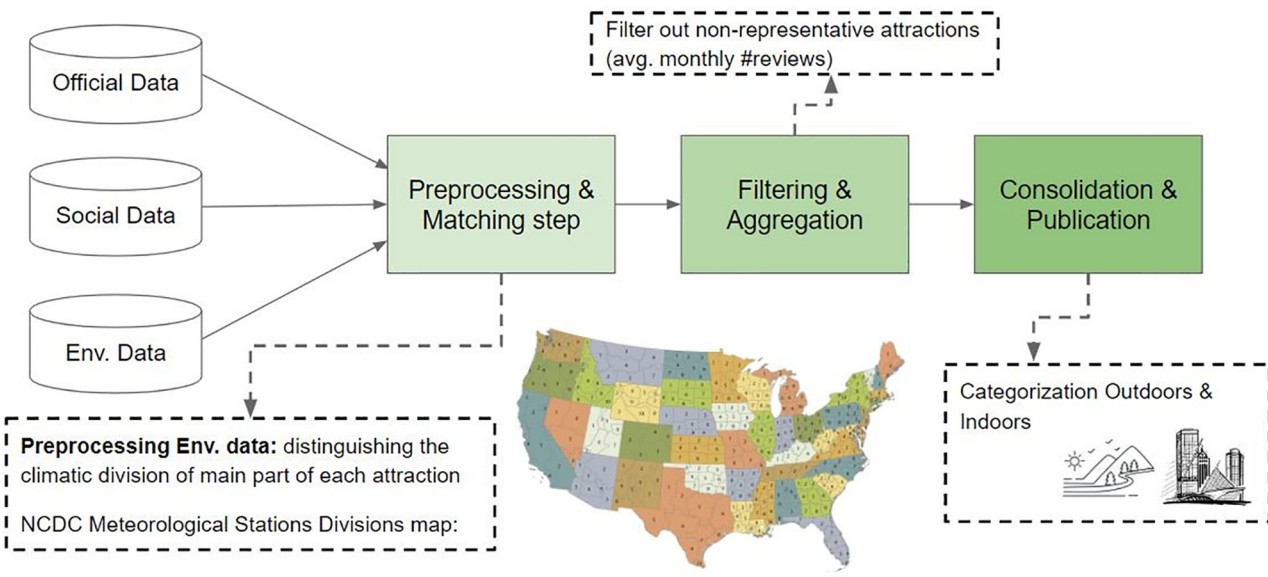

**Fig 1. Data collection phases.**

average temperature, visits in the last month, visits in the last year) serve as the input of the prediction models. A time series $X_i$ is a sequence $\{x_i^{(1)}, x_i^{(2)}, ..., x_i^{(t)}\}$, where $x_i^{(t)}$ denotes the value of variable $X_i$ measured (time-lagged) in time window $t$ for the specific touristic attraction that is the target of prediction. **Measured** variables are social media and environmental features that have been measured at timestamp $t$ while recency and seasonality features correspond to the history of visitation counts (i.e. response variable) in recent months or last year of visitation, which have been augmented and **time-lagged** to the time window $t$.

The objective function ($f$) is forecasting $y^{(t)}$, the tourism visitation at timestamp $t$ in a target attraction with the lowest prediction error, giving the input vector $X$ as the feature-set including social media, environmental, recency and seasonality features for each time window in the interval of $[1, t-k]$ (for $k > 0$), i.e., $\{x_1^{(1)}, x_2^{(1)}, ..., x_m^{(1)}, x_1^{(2)}, x_2^{(2)}, ..., x_m^{(2)}, ..., x_1^{(t-k)}, x_2^{(t-k)}, ..., x_m^{(t-k)}\}$ where $m$ is the number of available features (in some cases the objective function f combines a input vector X and the response variable y).

**Table 2. Overview of indoors (I) and outdoors (O) datasets and features (VIS: visits, SOC: social media features, ENV: environmental features.**

| Dataset | Provider | Granularity | Features | Data Range |
|---|---|---|---|---|
| I.VIS | Department for Digital, Culture, Media and Sport of England | monthly | total number of visitors to museums and galleries | 2004-03 to 2018-07 |
| I.ENV | U.K national weather service (Met Office) | monthly | min and max temperature, rainfall, sunny hours and days of air frost | 1996-01 to 2018-08 |
| I.SOC | TripAdvisor travel website | monthly | number of reviews, average ratings | 2001-08 to 2018-08 |
| O.VIS | U.S. National Park Service | monthly | total number of visitors | 1996-01 to 2016-08 |
| O.ENV | U.S. National Climate Data Center | monthly | minimum, average, maximum temperature, average precipitation | 2000-01 to 2016-10 |
| O.SOC | TripAdvisor travel website | monthly | number of reviews, average ratings | 2011-01 to 2016-09 |

## 2.3 Prediction techniques

This section offers a brief description of the techniques exploited for forecasting fine-grained tourist visit counts. Building on top of our previous work—the current state-of-the-art in the field [5], the learning models with the best prediction accuracy were selected, namely Support Vector Regression (SVR) [29], and the Auto Regressive Integrated Moving Average (ARIMA) based models -Seasonal ARIMA (SARIMA) [8] and Seasonal ARIMA with eXogeneous variables (SARIMAX) [30]. In here a Deep Neural-Network based method—Long Short Term Memory (LSTM) [31]— is also considered, which has not been included in [5] and is currently a popular method. Finally, we introduce two naive models in which are simple models that are based exclusively on historical observation [32].

The objective of all techniques is to estimate $y^{(t)}$, the number of visits in a given touristic place in the timestamp $t$ giving the input vector $X$ as the feature-set including social media, environmental, recency and seasonality features for each time window in the interval of $[1, t-k]$. However, there are some variations in the way each model adopts the features. For instance SARIMA models can exploit only the history of the number of visits, while SARIMAX exploits not only such history, but also the complete feature-set. The other models use the complete feature set, i.e. social media, environmental, recency and seasonality features.

**2.3.1 Support Vector Regression.** Support Vector Regression (SVR) is an extension of Support Vector Machines (SVM) widely used for regression tasks [29]. SVR performs a "linear regression" in a high-dimensional feature space resulting from a (nonlinear) mapping provided by a kernel function. The linear model (in the feature space) is given by:

$$f(X, W) = \sum_{j=1}^{m} W_j g_j(X) + b, \tag{1}$$

where $W$ is the weight vector to be "learned", $g_j(X)$ denotes a set of nonlinear transformations on the input feature set, and $b$ is the "bias" term. SVR pursues the best trade-off between the model's empirical error and the model complexity by constraining SVR regression function f(,) to the hyper-planes function class, and employing a margin around the hyper-plane. Moreover, f(,) only depends on a reduced set of the training data called the Support Vectors (SV), those which correspond to the active constraints in the optimization problem [29] defined as:

$$L(y, f(X, W)) = \begin{cases} 0 & if |y - f(X, W)| \leq \epsilon \\ |y - f(X, W)| - \epsilon & otherwise \end{cases} \tag{2}$$

where $y$ is the value to estimate.

The key parameters of SVR are the kernel function $K$, the margin of tolerance $\epsilon$, and the trade-off $C$ between the model complexity and the degree to which deviations larger than $\epsilon$ are tolerated.

**2.3.2 Seasonal Auto Regressive Integrated Moving Average (SARIMA).** Auto Regressive Integrated Moving Average models (ARIMA), is a classical time series forecasting method which was firstly proposed by Box and Jenkins [33]. In this model, the future value of a time series is a linear function of previous values of the original series and random errors. In other words, ARIMA projects the future values of a series based entirely on its own inertia. Thus, the set of predictor variables $X$ used by ARIMA consists of the past measurements of the response variable $y^{(t)}$, that is, $X = \{y^{(1)}, y^{(2)}, \ldots, y^{(t-k)}\}$, $k > 0$. When a seasonal effect is observed, a generalization of the ARIMA model is used, i.e. the SARIMA model. A SARIMA model is an

equation in the following form:

$$f(X, t) = \frac{\Theta \; \theta \; \epsilon^{(t)}}{\Phi \; \varphi \; \Delta \; \delta},$$

(3)

where $\Theta$, $\Phi$ and $\Delta$ are polynomials that compute the seasonal auto-regressive, differences and moving average components, respectively, $\theta$, $\varphi$ and $\delta$ quantify the respective regular (non seasonal) polynomials and $\epsilon^{(t)}$ is the estimation error.

**2.3.3 Seasonal Auto Regressive Integrated Moving Average with eXogenous variables (SARIMAX).** Due to the importance of exogenous data (i.e., social media, environmental data, recency and seasonality features) in our experiments, SARIMAX models (SARIMA with exogenous variables) [30] are applied. SARIMAX in addition to the history of response variable, takes into the account the input features, i.e. the external predictor variables, that is, the set of time series $X = \{x_i^{(1)}, x_i^{(2)}, ..., x_i^{(t)}\}$ where $x_i^{(t)}$ denotes the value of variable $X_i$ measured (time-lagged) in time window $t$. The SARIMAX model could be formulated as:

$$f(X, t) = \frac{\Theta \; \theta \; \epsilon^{(t)}}{\Phi \; \varphi \; \Delta \; \delta} + \beta X,$$

(4)

where the definition of the parameters $\Theta$, $\Phi$, $\Delta$, $\theta$, $\varphi$ and $\delta$ is as same as Eq 3.

**2.3.4 Neural network.** Introduced in [31], Long Short-Term Memory (LSTM) neural network models are well-suited to classification and regression as well as prediction tasks based on time series data. LSTMs have a notion of memory that may help capturing past trends in the data. The use of LSTMs in the context of tourism prediction is not new; in [34] the authors apply LSTM to tourism flow prediction, presenting interesting results.

A LSTM network consists of a chain of cells—each LSTM cell is configured by four gates: input gate, input modulation gate, forget gate and output gate. Input gates take new inputs from outside and process newly incoming data. Memory gates take inputs from the output of the LSTM cell in the last iteration. Forget gates decide when to forget the output results, thus selecting the optimal time lag for the input sequence. Output gates take all results calculated and generate final output [31].

Consider a time-series input represented as $X = \{x_i^{(1)}, x_i^{(2)}, ..., x_i^{(t)}\}$ where $x_i^{(t)}$ denotes the value of variable $X_i$ measured (time-lagged) in time window $t$ and hidden state cells $H = \{h^{(1)}, h^{(2)}, ..., h^{(t)}\}$ For $t = 1, ..., T$ LSTM computes:

$$f(X, t) = W_{hy} h^t + b_y$$

(5)

$$h_t = H(W_{hy} x^t + W_{hh} h^{t-1} + b_h),$$

(6)

where $W$ and $b$ are respectively weight matrices and bias vector parameters which need to be learned during model training.

**2.3.5 Naive models.** In general, naive forecasting models are simple models that are based exclusively on historical observation [32]. Our work defines two naive models as our baselines based on seasonality and on recency of tourism activities—Naive-Seasonality and Naive-Recency. For a naive prediction with seasonality, a simple approach to determine $y^{(t)}$ in a time window $t$, is to pick the number of visits at $y^{(t-12)}$ in the available past data. Similarly, for recency, the naive model predicts $y^{(t)}$ based on the number of visits at $y^{(t-1)}$.

$$f(t) = \begin{cases} y^{(t-12)} & \textit{naive seasonality} \\ y^{(t-1)} & \textit{naive recency} \end{cases}$$

(7)

**Table 3. Prediction techniques comparison.**

| Method | Exploit history of visits | Exploit social media and environmental features | Consider temporal dependency among data observations |
|---|---|---|---|
| SVM | ✗ | ✓ | No |
| SARIMAX | ✓ | ✓ | Yes |
| SARIMA | ✓ | ✗ | Yes |
| LSTM | ✓ | ✓ | Yes |
| Naive Models | ✓ | ✗ | Yes |

Table 3 lists all prediction techniques presented in this section, summarizing their main characteristics. A rich and diverse set of techniques is exploited, aiming at investigating how each of them performs in our target prediction problem.

## 2.4 Model learning and parameterization

Note that a specific model is learned (and later evaluated) for each attraction (for each park, museum or gallery), and thus, there is a different parameter choice for each of them. Nonetheless, for the sake of brevity, the values of the best parameters reported next are averages over all attractions. Parameter tuning for the models was performed as follows.

**2.4.1 Support Vector Regression.** For Support Vector Regression(SVR), the kernel function is set to "linear", because in preliminary experiments it produced the best results, besides being more efficient (lower execution time). The cost $C$ parameter was varied in the interval of $[2^{-5}, 2^{10}]$, and the best value varied for different attractions; however on average the best value was $C = 116$. The tolerance $\epsilon$ was tested in the range of $(0, 1)$ with steps of 0.1 and 0.3 was found to be the best value of $\epsilon$ (again on average across all attractions).

**2.4.2 Seasonal Auto Regressive Integrated Moving Average and Seasonal Auto Regressive Integrated Moving Average with eXogeneous variables.** Regarding Seasonal Auto Regressive Integrated Moving Average (SARIMA) and Seasonal Auto Regressive Integrated Moving Average with eXogeneous variables (SARIMAX) models, the forecast package in R (available at https://github.com/robjhyndman/forecast) was used in order to optimally find the best parameters (order of each polynomial) of the SARIMA model, as well as to find the seasonality pattern of the data.

**2.4.3 Long Short-Term Memory.** Related to Long Short-Term Memory(LSTM), different network architectures were explored, applying the ADAptive Moment estimation (ADAM) optimizer [35] for parameter optimization. ADAM is an adaptive learning rate optimization algorithm, designed specifically for training deep neural networks. Best results were obtained by: (i) normalizing all the variables in the range of (-1,1); (ii) using the mean-squared-error metric for the loss function; (iii) using a sequential model with one dense layer consisting 100 neurons using the Keras library in python (Keras is an open-source neural-network library written in Python); and (iv) the following setting: number of epochs was set to 1000, dropout to 0.2 and batch size of 30.

## 2.5 Prediction architecture

Fig 2 depicts the methodology, with the division of the datasets into training and test sets, having social media, environmental, recency and seasonality features and number of visits as the response variable (y). Recency features consist of visit counts in the previous last 4 months (y-1, y-2, y-3, y-4) and their log values (log y-1, log y-2, log y-3, log y-4) while seasonality features are the number of visits in the last year, same period, i.e. (y-12, y-13, y-14, y-15) and their log

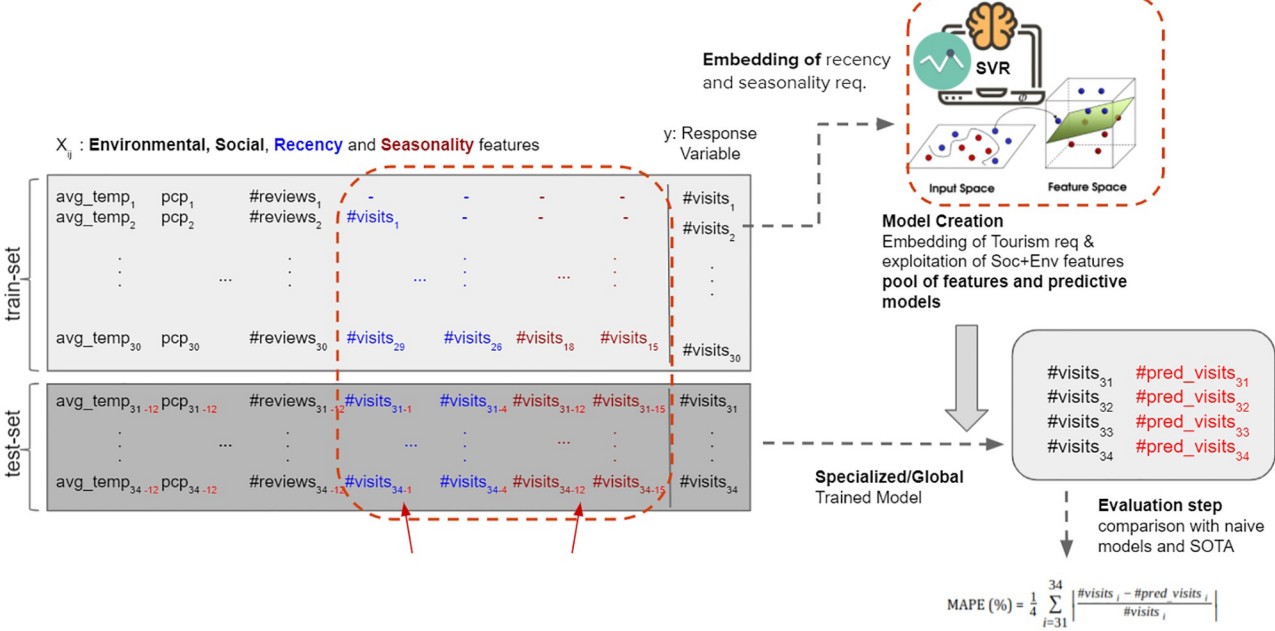

**Fig 2. Tourism demand prediction methodology adopting social media, environmental, recency and seasonality features.**

values (log y-12, log y-13, log y-14, log y-15). Furthermore, in our prediction architecture, as it can be seen in Fig 2, since social media and environmental data may not be available at prediction time, we exploit the values of the input feature in the last year ($X_{i-12}$) as the input of the models in the test case. This strategy has been used because of the annual seasonality behaviour of the tourism domain, as discussed in Section 1.

Fig 3 illustrates the construction of specialized and global models in order to study the prediction accuracy of the specialized models, with that of a global model trained with all attractions of each category of attractions (indoors our outdoors). Specialized models train a model particularly with features of each attraction while for global models, the model receives feature observations of all attractions of each type building a model.

In our experiments, cross-validation was performed to learn and optimize the prediction models. For each attraction, each time series was first divided into two parts: the **training set**, consisting of the first $m$ months of data ($m$ = 30 for outdoor attractions and $m$ = 76 in indoor attractions), and the **test set**, consisting of the remaining months of data (4 months for both outdoor and indoor attractions). The training set is used to learn the prediction model and optimize the model's parameters, while the test set is used for evaluating the learned model and reporting effectiveness results. For models requiring parameter tuning, the training set is further split randomly into additional parts.

**Cross-validation** with k = 10 (k-fold cross validation) was employed. In k-fold cross-validation, the training data is randomly partitioned into k equal sized sub-samples. Of the k sub-samples, a single sub-sample is retained as the validation data for testing the model, and the remaining k-1 sub-samples are used as training data. The cross-validation process is then repeated k times, with each of the k sub-samples used exactly once as the validation data. The k results can then be averaged to produce a single estimation. Note that as the process of choosing the validation sets is random, it can cause different models and model parameters (but very similar) in each execution and consequently slightly different prediction results.

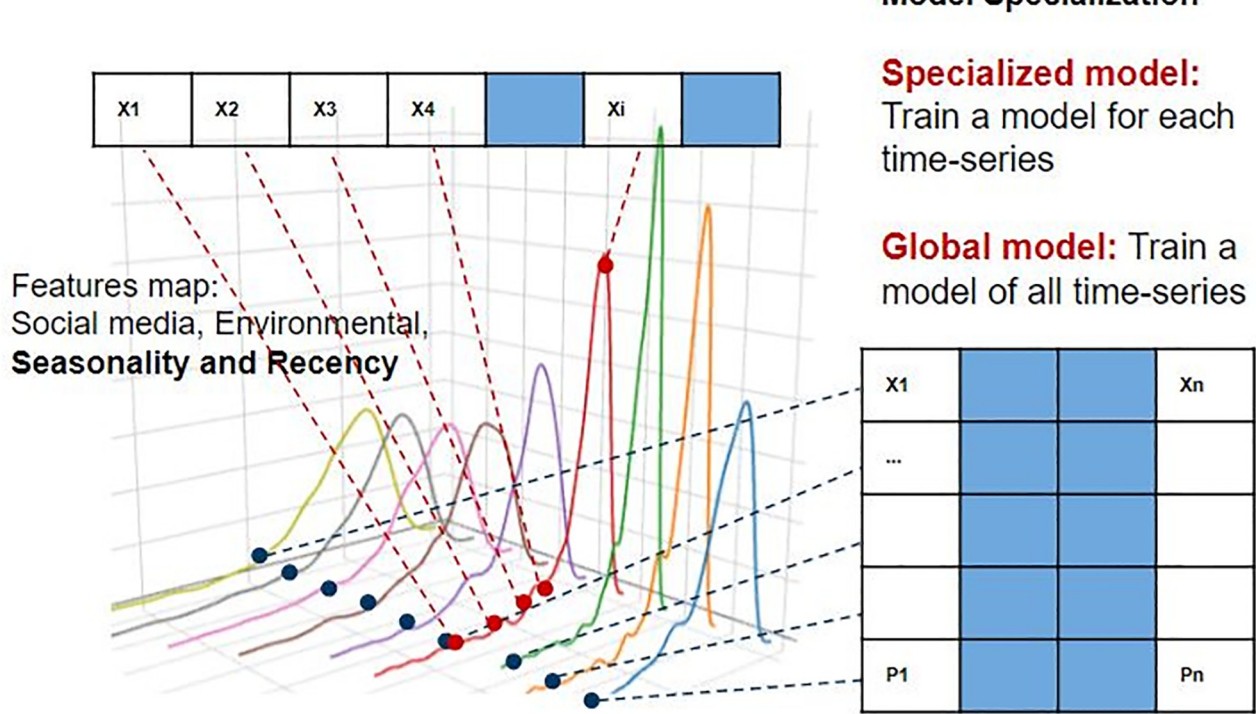

**Fig 3. Model specialization—illustration of global vs. specialized models.**

In order to evaluate the accuracy of the prediction techniques, the Mean Absolute Percentage Error (MAPE) [36] is used, being defined as:

$$MAPE(\%) = \frac{1}{M} \sum_{t=1}^{M} |\frac{y^{(t)} - \hat{y}^{(t)}}{y^{(t)}}| \tag{8}$$

where $M$ is the number of forecasting periods, $y^{(t)}$ is the actual visitation count and $\hat{y}^{(t)}$ is the predicted visitation count, both for time window $t$. A lower MAPE(%) value indicates a smaller percentage of errors produced by the prediction model. One commonly used interpretation of MAPE(%) values was suggested by [36] as follows: less than 10% is highly accurate forecasting, 10%-25% is good forecasting, 25%-50% is reasonable forecasting, and 50% or more is inaccurate forecasting.

## 2.6 Factorial design of tourism key requirements

To further investigate how recency, seasonality and model specialization requirements impact the prediction accuracy of different techniques, a factorial design analysis is performed over the correspondent features of each requirement to quantify the relative importance of each individual feature as well as their interactions on prediction accuracy.

A $2^k$ experimental design technique was employed, since we are interested in determining the effect of $k$ factors, each of which having two alternatives or levels. Such a design can be analyzed using a regression model to compute the main effect of a given factor $x_i$, subtract the average response of all experimental runs for which $x_i$ was at its low (False) level from the

average response of all experimental runs for which $x_i$ was at its high (True) level [15]. The importance of a factor is measured by the proportion of the total variation in the response variable that is explained by this factor.

Specifically, a $2^k$ factorial design was employed with $k = 3$ factors (i.e.,recency, seasonality and model specialization), each one with two levels (true or false). This design allows us to estimate the relative importance of each factor as well as all factor interactions on the response variable. This importance is estimated by the fraction of the total variation observed in the response that can be explained by each factor (or factor interactions). In the following, we define the considered factors and factor levels. Note that a $2^k$ factorial analysis is performed for each type of attraction (indoors and outdoors):

- Recency factor ($\mathcal{R}$): two levels are defined: (1) True, if the visit counts in the previous last 4 months (y-1, y-2, y-3, y-4) and their log values (log y-1, log y-2, log y-3, log y-4) are used for training the model and; (2) False, otherwise.

- Seasonality factor ($\mathcal{S}$): two levels are defined: (1) True, if the visit counts in last year (y-12, y-13, y-14, y-15) and their log values (log y-12, log y-13, log y-14, log y-15) are used for training the model and; (2) False, otherwise.

- Model Specialization factor ($\mathcal{M}$): two levels of are defined: (1) True, if an individual model for each indoor/outdoor venue is trained and; (2) False, a unique model for all venues of each attraction class is learned.

## 3 Experimental results

Our analysis starts by introducing the baselines for both indoors and outdoors attractions. Then, we discuss the impact on the results of the incorporation of the two tourism requirements—seasonality and recency—as features into the current SOTA prediction model (from [5]). RQ1 is answered at the end of these two sections.

In Section 3.3, RQ2 is investigated by applying a factorial design analysis in order to evaluate the impact of each factor (requirements) as well as of their interactions on the prediction accuracy of the models. A finer evaluation of the impact of each of the features is provided by an analysis of the coefficients of those features in the final models (Section 3.4).

In order to answer RQ3, the impact on prediction accuracy of each of the recency and seasonality features in different scenarios of scarcity of historical data is analyzed (Section 3.5). Finally, in Section 3.6, some examples of attraction are presented for which only by exploiting the complete set of tourism requirements, good prediction results could be obtained.

### 3.1 Comparison among baselines

In our previous work [5], we provided evidence of the importance of considering model specialization as an explicit requirement for tourism prediction. Our proposed prediction model —a specialized SVR method adopting social media and climate data—outperformed alternative models in the literature. We also compared the prediction accuracy of the specialized SVR models trained separately for each attraction with that of a global SVR model trained with all attractions of each type—indoors and outdoors.

In this section, for the sake of self-contention, the main results from [5] are summarized, adding to those results a comparison with a new method not exploited in that work—a Long Short-Term Memory (LSTM) neural network model, a popular method was used as as baseline for comparison. For the sake of completeness, our analysis also includes naive and classical models for recency and seasonality, also as baselines. The naive models are included since they

directly reflect the behaviour of recency and seasonality features without any additional information in their predictions. Finally, considering the great popularity of the Neural Network models in the recent years, our analysis included one of the most successful neural network models in the domain of time-series and tourism prediction [31] which is LSTM. In the following, we elaborate on the baselines and their respective results.

**3.1.1 Naive models.**   Two naive models are defined based on seasonality and on recency of tourism activities—Naive-Seasonality and Naive-Recency. For a naive prediction with seasonality, a simple approach to determine $y^{(t)}$ in a time window $t$, is to pick the number of visits at $y^{(t-12)}$ in the available past data. Similarly, for recency, $y^{(t)}$ is predicted based on the number of visits at $y^{(t-1)}$.

**3.1.2 Classical Models.**   ARIMA models are one of the classical time-series prediction techniques widely used in the tourism prediction task. Accordingly, results of SARIMAX and SARIMA models are also reported where Seasonal-ARIMA (SARIMA) incorporates the known seasonality (periodicity) of the data into an ARIMA model, enhancing its predictive power. The SARIMAX baseline adds social media and climate features as the exogenous features. Both models were considered in [5], however for the sake of completeness, those results are replicated in this paper.

**3.1.3 State-of-the-art (SOTA) model.**   In our previous work [5], we observed that, training specialized models (spe.) for each attraction using SVR outperformed the case where a single global model (glo.) is trained for all attractions of each type. We report both results (SVR spe. and SVR glo.) alongside the other baselines.

**3.1.4 Neural network model.**   In here a robust neural network model—LSTM— which is actively used in time-series prediction tasks, is also exploited. LSTM models are famous for their memory-based architecture capable of capturing past trends in the data.

Accuracy results for each of these models are presented in Tables 4 and 5 for indoor and outdoor attractions, respectively. Since our goals is to predict the number of visits with the best possible accuracy, we focus our attention on the cases where MAPE is lower than 25% (MAPE < 25%—accurate predictions). In this scenario, the specialized SVR model with Environmental and Social features is the best model, predicting accurately for the highest percentage of attractions (almost 93% of Museums and 95% of Parks), considerably outperforming other models. However, best results for MAPE less than 10% (highly accurate results) are achieved by the naive-recency (26%) for the indoor attractions and by the naive-seasonality (42%) for the outdoor attractions. The success of the naive methods in highly accurate results (MAPE < 10%) is one of the reasons that motivates the adoption of recency and seasonality in

**Table 4. Baseline results for 27 museums in U.K. (indoors).** The values in the table represent the percentage of attractions with MAPE in each specified range.

| | Museums | | | | | | |
|---|---|---|---|---|---|---|---|
| **MAPE** | **naive recency** | **naive seasonality** | **SARIMAX** | **SARIMA** | **LSTM** | **SVR spe.** | **SVR glo.** |
| MAPE<10 | **25.93%** | 18.51% | 11.11% | 11.11% | 11.11% | 14.81% | 3.7% |
| MAPE<25 | 70.37% | 81.48% | 85.19% | 74.07% | 74.07% | **92.59%** | 11.11% |

**Table 5. Baseline results for 76 national parks in U.S. (outdoors).** The values in the table represent the percentage of attractions with MAPE in each specified range.

| | Parks | | | | | | |
|---|---|---|---|---|---|---|---|
| **MAPE** | **naive recency** | **naive seasonality** | **SARIMAX** | **SARIMA** | **LSTM** | **SVR spe.** | **SVR glo.** |
| MAPE<10 | 6.58% | **42.11%** | 13.16% | 7.89% | 23.68% | 22.37% | 5.26% |
| MAPE<25 | 56.58% | 82.89% | 69.74% | 39.47% | 84.21% | **94.74%** | 18.42% |

our specialized models. The success of naive seasonality in parks is associated with the seasonal-cyclic behavior of climate in this type of outdoor attractions, as seasonality has been considered one of the main phenomena affecting tourism, principally due to changes in the weather conditions [9]. Climate conditions are less important when inspecting indoor attractions such as museums and galleries. (Naive) Recency, on its turn, in the case of indoor sites demonstrated to be a very good predictor for specific cases. Next, the performance of recency and seasonality is evaluated when explicitly incorporated as features into the specialized SVR model.

**3.1.5 Computational complexity and execution time.** The computational complexity of the Support Vector Machine (SVR) for time complexity is of $O(N^3)$ and for space complexity is of $O(N^2)$ where N is the number of points [37]. In our experiments, as few training points (N = 30 for outdoor attractions and N = 76 in indoor attractions) are used and the model is trained once a month, the execution time for the prediction task is not a major concern.

The computational complexity of other algorithms is as following: for SARIMA models the complexity is in the order of $O(n)$ while in the case of neural networks they take more time due to numerous iterations applying forward and back-propagation—back-propagation, in the order of $O(n^5)$ is much slower than the forward propagation, in the order of $O(n^4)$. Finally, for the case of naive models the complexity is $O(1)$ since they only pick the defined index of historical data to pass as the naive prediction.

All in all, for specialized SVR models, the mean execution time is about 45 seconds (min exec. time: 21 seconds and max exec. time: 92 seconds) while for generalized models for different sets of features the mean execution time was around 4 hours (min exec. time: 1 hour and 33 minutes and max exec. time: 20 hours). After conclusion of the training step, the prediction phase is quite fast—average of 4 seconds for all 100 attractions independent of global or specialized models. The machine used in our experiments was a desktop PC with 4 CPUs and 16 GBs of RAM memory using the R programming language.

## 3.2 Augmentation with RECENCY and SEASONALITY

Our focus now shifts to demonstrate how the addition of the other two tourism requirements, i.e. recency and seasonality, into the state-of-the-art specialized SVR models with Social and Environmental features, hereafter called SpecES (Specialized with Environmental and Social)), can improve the accuracy of forecasting the visitations for fine-grained touristic points. Results of adding recency and seasonality to global models can be found in the Appendix A. Training specialized models for each individual attraction allows the models to learn specific patterns of visitation at each touristic point. Tables 6 and 7 show the prediction performance of the models when all the three tourism prediction requirements are present i.e. model specialization, seasonality and/or recency. As previously discussed, for indoor attractions, the specialized models without the new features (SpecES) have a good performance (MAPE < 25%)—over 92% for museums and 95% for parks (column *SpecES* in Tables 6 and 7 refer to column *SVR. spec* in Tables 4 and 5, repeated here to facilitate comparison).

**Table 6. SpecES prediction results augmented with the other two tourism requirements—seasonality and/or recency, trained for each of the 27 museums in U.K. (indoors).** The best prediction models are in bold face.

| | Museums | | | |
|---|---|---|---|---|
| **MAPE** | **SpecES** | **SpecES+recency** | **SpecES+seasonality** | **SpecES+recency+seasonality** |
| MAPE<10 | 14.81% | 29.63% | **48.15%** | **48.15%** |
| MAPE<25 | 92.59% | **96.30%** | **96.30%** | 92.59% |

**Table 7. SpecES prediction results augmented with the other two tourism requirements—Seasonality and/or recency trained for each of the 76 national parks in U. S. (outdoors).** The best prediction models are in bold face.

| | Parks | | | |
|---|---|---|---|---|
| MAPE | SpecES | SpecES+recency | SpecES+seasonality | SpecES+recency+seasonality |
| MAPE<10 | 22.37% | 40.79% | 48.68% | **50.00%** |
| MAPE<25 | 94.74% | 94.74% | **96.05%** | **96.05%** |

Considering the results in Tables 6 and 7, it can be noted that the combination of only seasonality and model specialization for museums (fourth column in Table 6) results in a slightly higher accuracy (96% for MAPE < 25 and 48% for MAPE < 10) than when all features are used. For parks, instead, the combination of all tourism requirements (fifth column in Table 7) performs the best (96% for MAPE < 25 and 50% for MAPE < 10). This aspect will be further analyzed in the next section when we perform a factorial analysis over the tourism requirements.

Regarding the high accuracy cases (MAPE < 10), remind that the combination of SpecES with the other two key tourism requirements- recency and seasonality- produced the best overall results. In more details, for the indoor attractions (Table 6), *SpecES+recency+seasonality* produced high prediction accuracy for about 48% of the museums compared to 22% obtained by the naive-recency (Table 4), the best baseline in this category. A similar behavior is seen for the outdoor attractions—comparing the results in Tables 5 and 7, for (MAPE < 10), *SpecES+recency+seasonality* has highly accurate predictions for 50% of the parks compared to around 42% using the naive-seasonality.

## 3.3 Factorial analysis

This section investigates the impact of each of tourism prediction requirements, i.e. recency, seasonality and model specialization by means of a factorial design analysis. Factorial design techniques help to analyze the effect of each factor (requirement) as well as the effects of their interactions on the tourism demand (visits count) in each touristic attraction.

We employ a regression analysis for evaluating the amount of variation in the prediction results that can be explained by each factor (and interaction). A $2^k r$ experimental design technique was adopted to estimate the effect of $k = 3$ factors (recency, seasonality and model specialization), each of which having two levels (requirement is incorporated into the model or not, for the prediction task) and with $r$ replications per configuration. As reported in Section 2.3, applying cross-validation along with the SVR model produces small variations in prediction results due to the stochastic nature of the task. In order to reduce this variation and increase the accuracy of results, each experiment was executed several times to calculate the average and standard deviation of the variation of results. The adequate number of runs was estimated based on 95% confidence level and accepted error percentage of 2%, as being 5 runs. In our factorial analysis, the response variable is the % of attractions (indoors/outdoors) that fall in each MAPE range. The goal is to estimate the importance of each factor (interaction) on the variation observed in those % of touristic attractions. When all three requirements are turned off, the global SVR model (non-specialized model trained for all attractions of each type—indoors and outdoors) is used with only the Environmental and Social media features, i.e. absence of all three factors. Results of adding recency and seasonality to global models can be found in the Appendix B in Tables 14 and 15.

Table 8 shows the variation explained by each tourism requirement on the prediction results in each category of attractions. It can be observed that in both indoors and outdoors

**Table 8. Contribution of each of tourism prediction requirements: Recency, seasonality, model specialization and their interactions into the response variable in each category of attractions: Parks and museums; results for MAPE < 10 and MAPE < 25 in 5 runs.** The contributions higher than 5% are in bold face. The analysis of the statistical significance of our numerical results are presented in details in the Appendix B in Tables 16 and 17.

| Requirements | contribution (%) | | | |
| --- | --- | --- | --- | --- |
| | MAPE < 25 | | MAPE < 10 | |
| | Museums | Parks | Museums | Parks |
| Recency | 1.0 | 1.1 | 0 | 1.1 |
| Seasonality | **21.1** | **24.6** | **19.8** | **32.6** |
| Model spec. | **55.9** | **46.0** | **70.0** | **58.5** |
| Recency, Seasonality | **5.5** | 0.7 | 1.8 | 2.0 |
| Recency, Model spec. | 0.1 | 1.2 | 1.3 | 2.0 |
| Seasonality, Model spec. | **13.3** | **24.4** | 2.9 | 1.8 |
| Recency, Seasonality, Model spec. | 0.7 | 1.8 | 1.6 | 0.9 |
| Residuals | 2 | 0.2 | 3 | 1 |

attractions, model specialization and then seasonality have the largest contributions. In the case of MAPE < 25, there is also a significant contribution of the interaction between these two factors—seasonality and Model specialization (24.4% for Parks and 13.3% in the case of Museums). The analysis of the statistical significance of our numerical results are presented in details in the Appendix B in Tables 16 and 17.

In addition, it can be observed that model specialization, in relative terms, is more important to the variation observed for MAPE < 10 than for the results for MAPE < 25 (explains 70% versus 56% of result variation for Museums and 59% versus 46% for Parks). One may say that if highly accurate prediction results (MAPE < 10%) are needed, the use of specialized models becomes even more important.

Table 8 also shows that the impact of recency and its interactions with other factors on the prediction results are almost negligible. Despite that, recency can improve results (look for instance at the second and third columns in Tables 6 and 7), indicating that it should be used, mainly if the seasonality features are not available.

Seasonality alone has more than 20% of contribution in both parks and museums, for MAPE < 25. This indicates that when only the historical data for an attraction is present, significant improvements in accuracy can be obtained by injecting seasonality features into the model as input variables. It can also be observed that seasonality has even a higher impact (32.6% versus 19.8%) in outdoor attractions for very accurate prediction results (MAPE < 10). This is in alignment with what was discovered in [5] when we showed that in outdoor attractions the impact of climate features is much higher than in indoor attractions, considering that the climate features have a high correlation with seasonality [10].

## 3.4 A drill down analysis of encapsulated features in recency and seasonality factors

In the previous section, the impact of each of the tourism prediction requirements was quantified. In the following, we delve further into the role that each of the recency and seasonality features (introduced in the Section 2) play regarding the prediction task accuracy. We will do so by analyzing the learned coefficients of the **global models** in the indoor and outdoor scenarios. In other words, the global models will be used as an **analytical tool (only)**. This option was chosen to avoid the complexity of analysing all the 103 models produced with specialization (one for each attraction).

**Table 9. The coefficients of features of global (single) model for all 27 U.K museums adopting each time a different set of features: (I) only social media and environmental features (soc+env), (II) social media, environmental and recency features (soc+env+rec), (III) social media, environmental and seasonality features (soc+env+seas), (IV) complete feature set: Social media, environmental, seasonality and recency feature (soc+env+rec+seas).** The **bold** face shows the top 2 features in each column.

| Features | soc+env | soc+env+rec | soc+env+seas | soc+env+rec+seas |
|---|---|---|---|---|
| tmin | -0.093 | 0.025 | -0.004 | -0.031 |
| tavg | 0.024 | 0.005 | -0.001 | 0.001 |
| tmax | **0.116** | -0.011 | 0.002 | 0.026 |
| air_frost_days | 0.004 | 0.005 | 0.000 | 0.008 |
| rain | -0.022 | 0.006 | 0.003 | 0.018 |
| sunny_hr | -0.037 | 0.020 | 0.004 | 0.022 |
| revs | **0.517** | 0.010 | 0.004 | 0.002 |
| rating | -0.051 | -0.001 | 0.000 | -0.004 |
| month | -0.007 | -0.060 | -0.002 | -0.026 |
| y-1 | - | **0.511** | - | **0.407** |
| y-2 | - | **0.258** | - | 0.158 |
| y-3 | - | 0.156 | - | 0.033 |
| y-4 | - | 0.054 | - | 0.026 |
| log y-1 | - | 0.025 | - | 0.089 |
| log y-2 | - | -0.003 | - | -0.033 |
| log y-3 | - | -0.032 | - | -0.015 |
| log y-4 | - | 0.003 | - | -0.017 |
| y-12 | - | - | **0.764** | **0.658** |
| y-13 | - | - | 0.038 | -0.291 |
| y-14 | - | - | 0.084 | -0.051 |
| y-15 | - | - | **0.089** | 0.034 |
| log y-12 | - | - | 0.011 | -0.028 |
| log y-13 | - | - | -0.002 | -0.027 |
| log y-14 | - | - | -0.003 | 0.036 |
| log y-15 | - | - | -0.007 | 0.003 |

As can be seen in Tables 14 (indoors) and 15 (outdoors) (in the Appendix A for the sake of space, easiness of analysis and flow of discourse), the impact of the incorporation of the recency and seasonality features into the global models is similar to that of the specialized models, with significant improvements over the case in which such features are not used, for MAPE < 10 and MAPE < 25, although results are not as good as with the latter.

Tables 9 and 10 show the learned coefficients of global models in indoor and outdoor scenarios, respectively. In more details, for this analysis, global models were built for all attractions of each type, adopting each time a different feature-set: (I) soc + env: global model trained having only social media and environmental features in the feature-set; (II) recency (soc + env + rec): global model having recency features in addition to the social media and environmental features; (III) seasonality (soc + env + seas): global model having seasonality features in addition to the social media and environmental features and; (IV) seasonality+recency (soc + env + rec) + seas): global model having all features including social media, environmental, recency and seasonality features.

A similar pattern can be seen in the learned coefficients of outdoor attractions (Table 10). The model has larger weights for average temperature and number of reviews in the simple model; y-1 and average temperature in the recency model; y-12 and y-14 in the seasonality

**Table 10. The coefficients of features of global (single) model for all 76 U.S. National Parks adopting each time a different set of features: (I) only social media and environmental features (soc+env), (II) social media, environmental and recency features (soc+env+rec), (III) social media, environmental and seasonality features (soc+env+seas), (IV) complete feature set: Social media, environmental, seasonality and recency feature (soc+env+rec+seas).** The bold face shows the top 2 features in each column.

| Features | soc+env | soc+env+rec | soc+env+seas | soc+env+rec+seas |
|----------|---------|-------------|--------------|------------------|
| tmin | 0.006 | -0.290 | -0.001 | -0.011 |
| tavg | **0.021** | **0.580** | 0.002 | 0.022 |
| tmax | 0.002 | -0.278 | 0.000 | -0.010 |
| temp_dif | -0.015 | 0.003 | 0.003 | 0.003 |
| pcp(rain) | 0.000 | -0.002 | -0.003 | -0.001 |
| revs | **0.278** | 0.019 | 0.001 | 0.000 |
| rating | -0.007 | 0.005 | 0.009 | 0.005 |
| month | -0.007 | -0.038 | -0.001 | -0.005 |
| y-1 | - | **1.218** | - | **0.340** |
| y-2 | - | -0.297 | - | 0.014 |
| y-3 | - | -0.070 | - | 0.018 |
| y-4 | - | 0.066 | - | 0.014 |
| log y-1 | - | -0.007 | - | 0.034 |
| log y-2 | - | 0.023 | - | 0.004 |
| log y-3 | - | -0.023 | - | -0.013 |
| log y-4 | - | 0.001 | - | -0.001 |
| y-12 | - | - | **0.947** | **0.928** |
| y-13 | - | - | 0.026 | -0.262 |
| y-14 | - | - | **0.031** | -0.018 |
| y-15 | - | - | -0.010 | -0.038 |
| log y-12 | - | - | 0.009 | -0.013 |
| log y-13 | - | - | -0.003 | -0.024 |
| log y-14 | - | - | -0.012 | -0.005 |
| log y-15 | - | - | 0.003 | 0.013 |

model; and finally y-12 and y-1 for complete feature-set model, which is consistent with our previous discussions in the factorial design analysis.

Regarding indoor attractions (Table 9), the learned coefficients indicate the high importance of number of reviews and then maximum temperature in the simplest model. In the recency model (soc + env + rec), instead, higher weights are given to the number of visits in the last two months (y-1 and y-2 features). The number of visits in the last year (y-12) and in 15 months before (y-15) are more relevant when seasonality is incorporated into the model (soc + env + seas). Finally, visits in the last year and in the last month (y-12 and y-1) contribute more to the accuracy of the complete model (soc + env + rec + seas). Interestingly, the impact of visits in the last year, same period (y-12) has a larger weight than visits in the last month (y-1) which is aligned with what was observed in the factorial analysis of the impact of tourism prediction requirements—seasonal features have more contribution to the model than recency ones.

## 3.5 Impact of historical data scarcity on the prediction task

As discussed in the previous sections, learning specialized models trained with the complete information regarding social, environmental, recency, and seasonality information considerably improves the accuracy of the prediction models. However, having full information

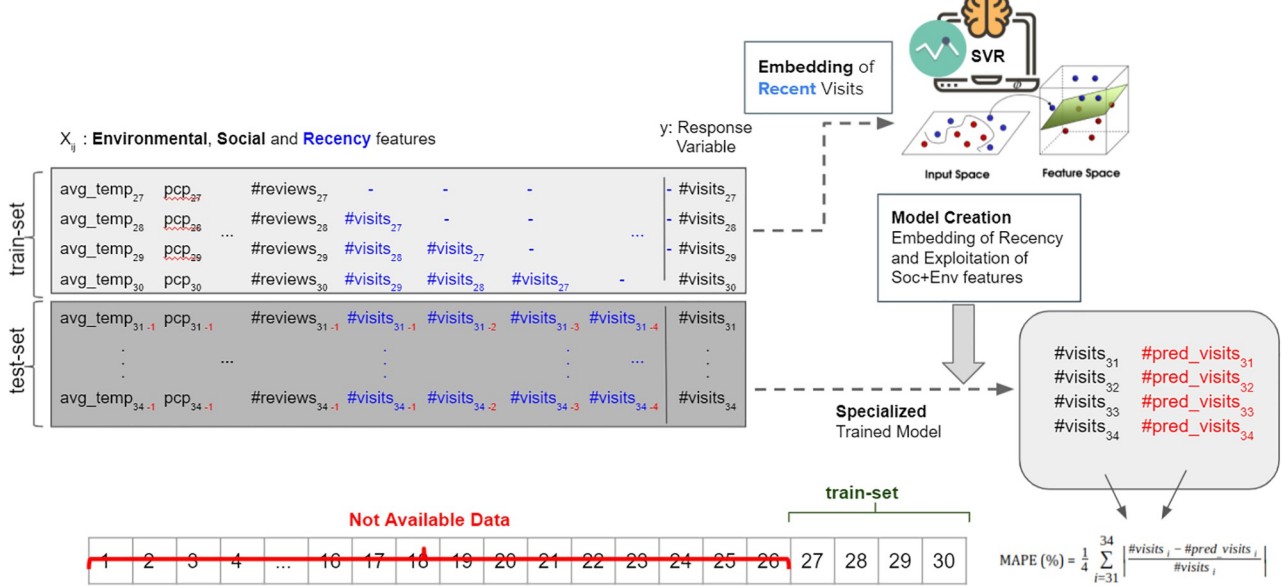

**Fig 4. Tourism demand prediction methodology in scarcity of seasonal data adopting social media, environmental and recency features.**

regarding recency and seasonality is not always guaranteed. In the following, we further investigate the individual impact of recency and seasonality in the prediction task in scenarios without full availability of historical information on (number of) visits, social media and environmental data for touristic attractions. For these analyses, we revisit the prediction architecture and redefine the training and test sets when necessary.

**3.5.1 Only recency—scarcity in seasonal data.** In scenarios in which there is not enough historical information for an attraction, i.e., there is only very recent data on visits, social media and environmental data of a touristic place, the model can exploit recency features in order to improve the prediction of the future visitation. This situation may occur, for instance, for new attractions or attractions that have only started to collect (visitation) data very recently. To simulate this scenario in our datasets, the model only uses the last four (4) months of the historical data of each attraction to train each prediction model while filtering out the rest of the data. Fig 4 presents our revised prediction architecture to deal with this new prediction scenario.

Since the features of the last 12 months are not available to evaluate the prediction model, we adopted two different scenarios for defining the input value of each feature in the test-set: (i) *last month* case, in which the previous month information is used as the input of the model and; (ii) *mean of 4-months* case, in which the mean of each feature of the train-set is used as the input feature values of the models. Tables 11 and 12 (two leftmost columns) show the results. The percentage of parks with an accurate prediction (MAPE<10) is quite low in both cases (about 2%) while the percentages are a little higher ($\approx$ 15%) in museums. Regarding

**Table 11. Scarcity in seasonal historical and recent data—Evaluation of performance of recency and seasonality features in 27 Museums in U.K.**

| MAPE | Only Recency | | Only Seasonality | |
|---|---|---|---|---|
| | last month case | mean of 4-months case | unavailable last 4 months | unavailable last year |
| MAPE<10 | 14.81% | 14.81% | 44.44% | 29.63% |
| MAPE<25 | 37.00% | 37.00% | 81.48% | 74.00% |

**Table 12. Scarcity in seasonal historical and recent data—Evaluating performance of recency and seasonality features in 76 national parks in U.S.**

| MAPE | Only Recency | | Only Seasonality | |
|---|---|---|---|---|
| | last month | mean of 4-months | unavailable last 4 months | unavailable last year |
| MAPE<10 | 1.32% | 2.63% | 43.00% | 0.00% |
| MAPE<25 | 25.00% | 21.00% | 85.00% | 21.00% |

good predictions (i.e., MAPE<25), using the last month as the input has the same results as using the mean of 4-months features (37% in museums in both cases) whereas regarding the parks, using the last month as the input has a slightly better performance(25%) than using the mean of 4-months features (21%).

**3.5.2 Only seasonality—scarcity in recent data.** Likewise the recency features, we analyze the performance of seasonality features when the most recent data is not available. This may happen in cases when data collection is periodical (or seasonal) and lasts longer periods and the most recent data is not yet available for prediction. In this scenario, seasonality features can be exploited, i.e. number of visits, social media and environmental data in the previous years in order to predict the future visitation, if this information is available. Fig 5 shows our revised prediction architecture to deal with this prediction scenario.

For this, the most recent historical data of each attraction are not used and only the remaining historical data are used for training the prediction model. For constructing the training-set, two cases are defined regarding the unavailability of historical data: (i) unavailable history of the last 4 months of each feature; (ii) unavailable last 12 months (last year) of each feature. The first case corresponds to the situation where the previous last 4 months (y-1, y-2, y-3, y-4) are not available while the second case is when we do not have one complete cycle of historical data (annual seasonality) [11].

Tables 11 and 12 (two rightmost columns) present the results for indoor and outdoor attractions. The percentage of museums with an accurate prediction (MAPE<10) is much

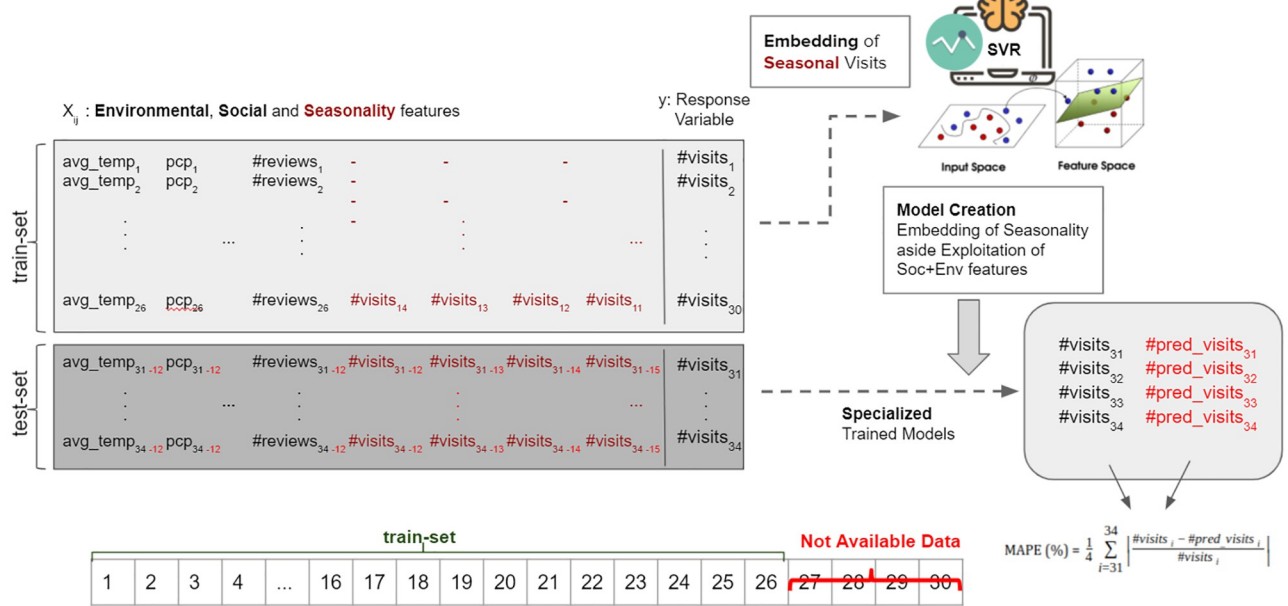

**Fig 5. Tourism demand prediction methodology in scarcity of recent data adopting social media, environmental and seasonality features.**

higher in the first case when only the last 4 months of the historical data is unavailable in comparison to the case when the complete historical data of the last year is missing (44% versus 30%). The scenario is even more dramatic for parks (43% versus 0%). This behaviour is similar for good predictions (MAPE<25) for both, indoor (81% versus 74%) and outdoor (85% versus 21%) attractions. These results again suggest the importance of the historical data. In other words, having the last trends of visitations besides the periodical/historical behaviors is essential for an accurate prediction.

### 3.6 Improving accuracy of difficult cases by incorporating explicit tourism prediction requirements

In our previous work [5], we have identified a small set of indoor and outdoor tourism attractions for which their best prediction models performed poorly. In this section, we evaluate whether the incorporation of seasonality and recency features into the specialized models for these attractions can help to mitigate the found problems. In particular, we focus on two attractions—National Portrait Gallery in U.K. and Bryce Canyon National Park in U.S.

In the case of National Portrait Gallery, the social media reviews had a non-typical major increase by April 2015 but there was a gradual decrease in the number of visits (Fig 6). This atypical behaviour could be explained considering the annual report published by National Portrait Gallery available at https://www.npg.org.uk/assets/files/pdf/accounts/npgaccounts2015—16.pdf), informing that the virtual audience grew on a national and international level during 2015/16 with an increased number of people having access to exhibitions, displays and the collection **online** through the gallery's website. As a result, more social media activity is observed but less in-site visitations. The incorporation of the recency and seasonality features helped to detect this behavior change and consequently improved the model accuracy (a reduction of 137% mean percentage error to 13% in Table 13).

Regarding the Bryce Canyon national park, the difficulty was that the considerable increase in number of visits (more than 20% starting in February 2016 until September of the same year in comparison with the same period in 2015) was not accompanied by social media reviews (same behavior as previous years plus a slightly decrease in May 2016 compared to May 2015) (Fig 7). A possible reason was the waiving of the entrance fees in 2016. Again, by

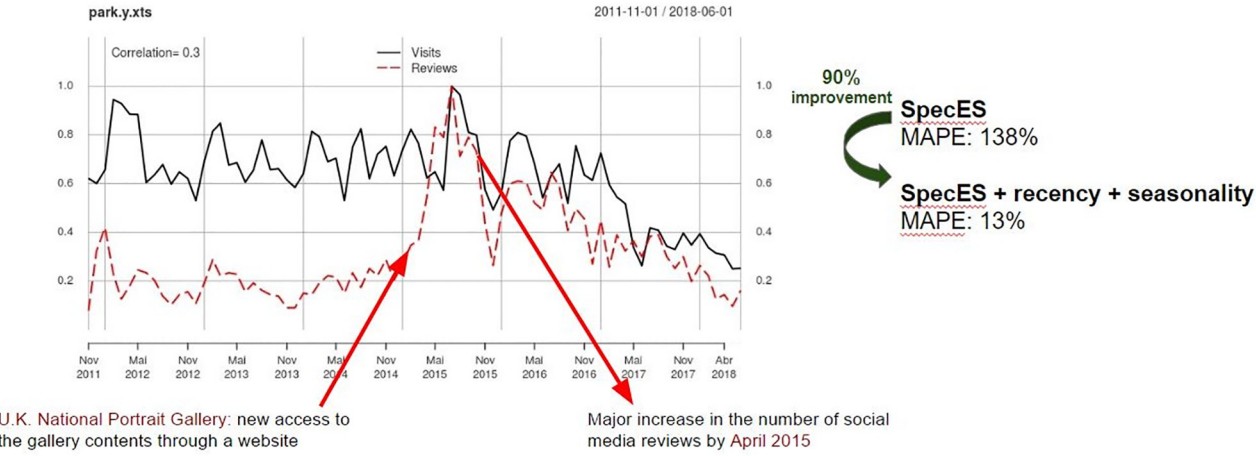

**Fig 6. Temporal evolution of number of visits and social media comments in National Portrait Gallery in U.K.—highly more accurate prediction model using specialized model with all recency and seasonality features.**

**Table 13. Accuracy of difficult cases incorporating explicit tourism prediction requirements in indoor and outdoor attractions.**

| Attraction | MAPE-SOTA results (from [5]) | MAPE—our results |
|---|---|---|
| U.K. National Portrait Gallery (indoor) | 137.90% | **13.34%** |
| U.S. Bryce Canyon National Park (outdoor) | 35.19% | **24.43%** |

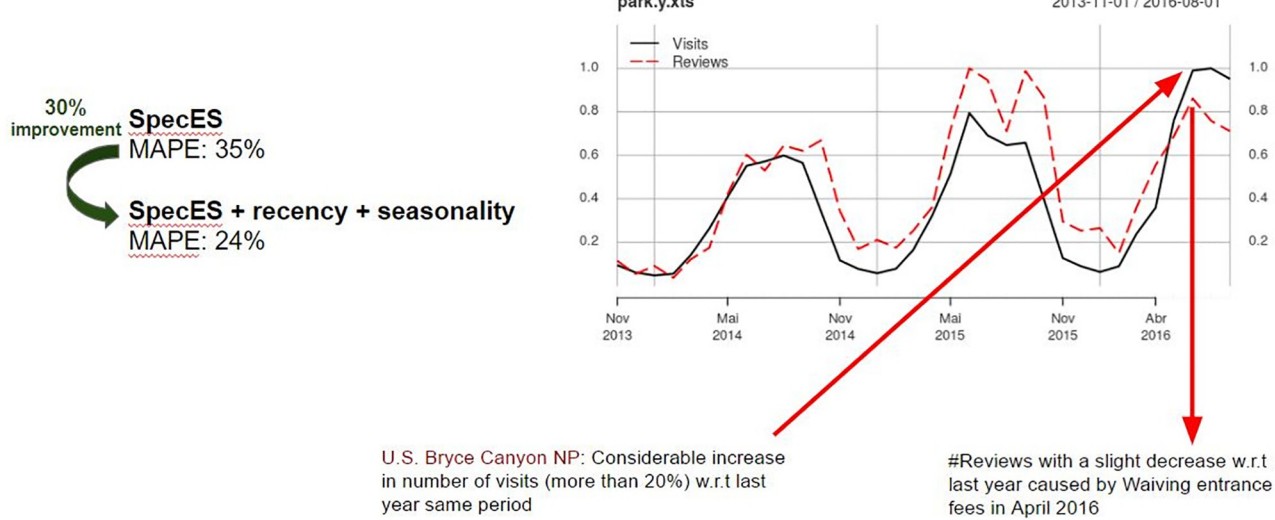

**Fig 7. Temporal evolution of number of visits and social media comments in Bryce Canyon National Park in U.S.—significant gain using specialized model with all recency and seasonality features.**

explicitly exploiting seasonality and recency such anomalies can be captured, reducing the mean percentage error of the models from 35% to 24% (reported in Table 13).

## 4 Discussion

### 4.1 Answering the posed research questions

This Section discusses answers to our posed Research Questions in Section 1.1 based on the experimental results presented in Section 3.

**RQ1** focused on analyzing whether key tourism prediction requirements could influence the prediction accuracy of fine-grained tourists' visits. Our experimental evaluation confirmed a positive answer to this question, corroborating our initial hypotheses. Significant gains have been observed over the previous state-of-the-art (SOTA) results [5], by explicitly incorporating the three requirements into the models.

In general, our analyses demonstrated a more prominent role of the seasonality features in the case of outdoor attractions (national parks), which can be explained by the weather´s seasonal behaviour that affects visitors´ decisions in paying a visit to an outdoor attractions. A significant importance was also given by the models to recency features, especially for the the case of indoor attractions (national museums). This fact can be justified by smaller importance of weather conditions in case of indoor attractions and a relatively significant impact of recent events such as festivals or pandemics in this type of attraction.

The impact of the incorporation of recency and seasonality as *explicit first-class features* into specialized models is perhaps better demonstrated by the capacity of the new models in

dealing with previously unsolvable cases by the SOTA. The National Portrait Gallery in the U. K., for instance, had a huge increase in social media reviews but that was not accompanied by real world visits, causing the SOTA models to mistakenly follow the social patterns, consequently resulting in low accuracy. The explicit incorporating of recency features helped the models to capture recent trends in visitations, diminishing the importance of the tendency observed for the social media.

Another example is the Bryce Canyon national park in the U.S., in which the visits experience some period of untypical increases. That increase was not explainable by neither the environmental features nor the social media reviews, solely inputs exploited by the SOTA model. In this example, the explicit incorporating of seasonality features helped the models to capture variations in cyclic behaviour of tourists, enhacing the prediction accuracy for this attraction.

**RQ2** aimed at quantifying the combined and isolated influence of each of key tourism prediction requirements in improving prediction accuracy. Usually, we observed that our specialized models with presence of recency and seasonal features gave a much higher importance for these features than for social or environmental features exploited by the SOTA models. Intuitively, **recency** features capture the impact of recent events on the prediction models that may deviate from the historical patterns. We have indeed observed that the incorporation of recent trends allowed small adjustments of the model, helping to avoid high drops in effectiveness. Let´s take again the hard-to-solve case of outdoor national park in U.S.—Bryce Canyon to illustrate our argument. An evaluation of features weights of the proposed model for this attraction showed relatively higher weights for recency (e.g. weight 0.74 for feature *y-1*: last month's number of visits) and environmental features (e.g., weight 0.018 of feature *tmax*: maximum temperature and weight 0.002 for feature *revs*: number of reviews). This resulted in a quick response to the deviations in visitations in the testing period, helping to produce a 30% reduction of prediction error compared to the SOTA model.

Similarly, **seasonality** features capture the inherently cyclic behaviour of tourism demands. Again, to exemplify, the explicit use of seasonality features was helpful to obtain more accurate predictions in the case of hard-to-solve case of National Portrait Gallery, UK (indoor). The specialized model trained for this attraction, could learn the seasonal cyclic behaviour of visitations over many years of historical data, not being mislead by the increase of visitations in a short period of time. In a drill down analysis of the features´ importance, a relatively higher weight was given to seasonal feature *y-12*: last year's number of visits (weight of 0.63) in comparison to social feature *revs*: number of reviews (weight of 0.46) that had been much important for the global model. As a consequence, our model could correct the gradual disassociation of social media review counts and number of visits, resulting in a model accuracy 9 times better than the SOTA model in the period of evaluation.

Finally, **model specialization** allowed us to capture very specific idiosyncratic patterns of visitations of individual parks that global models could not. For several attractions such as Aztec Ruins national park, the specialized model could obtain much more accurate results (up to 8 times better than global model). In these cases, particular recent and seasonal behaviors of visitations, alongside other social media and environmental features where better captured by the specialized model, not being confounded by general/global patterns of other attractions visitations' patterns. In other words, the weights of features assigned to each individual attraction, in our example Aztec Ruins national park, was better adjusted to the time-series of visitations for this particular attraction.

An interesting general pattern that deserves attention is the higher improvements for the high accuracy cases (MAPE < 10), mostly because the easier cases where solvable by the SOTA models, as explained above. Specialization, recency and seasonality stand out, as discussed

above, for the hard cases. But there still a lot of room for improvements, as we are still at a rate of 50% for this very accurate predictions.

To answer **RQ3**, we explored how scarcity in historical data—recent and seasonal features—impacted the prediction accuracy of models. This analysis aimed to simulate real world scenarios with lack of official census for some periods of time due to intermittent survey of visitation counts in touristic attractions. We observed that absence of recency or seasonality features drastically reduces the accuracy of prediction models. Although scarcity of recent data did not jeopardize the results as much as the absence of seasonality data did, recency features still had a significant impact on prediction accuracy, mainly for situations in which there was not enough historical data to capture seasonality for a given attraction.

This behaviour can be justified since for most touristic attractions, the seasonal touristic activities happen repetitively and repetition helps the learned model to emphasize certain behaviors as captured by the features. The lack of such seasonal data, consequently, may significantly reduce accuracy.

On the other hand, unusual or unexpected deviations from the historical touristic patterns because of recent events are rare and may not necessarily cause huge declines in prediction accuracy. These findings are inline with the good performance of the naive seasonal models.

## 4.2 Limitations and practical applications

Our analyses, though rich, have limitations. The main one relates to the lack of official data for some attractions in order to test our methodology on even finer time-grained data (weekly or daily basis). Predictions were evaluated on a monthly basis. This granularity of time was selected since the official ground-truth data was available and aggregated at this level. However some preliminary experiments suggest that there is a strong possibility of successfully applying the same methodology on a finer granularity of time.

Regarding the practical application of our results in real-word scenarios, although exploiting data from social media is fascinating, especially recent and seasonal data, a critical question that will determine their utility for forecasting future visitation is: how well do they reflect on-the-ground visitor surveys and records? In our work, we showed that there is a strong relationship between the number of reviews and visitation field-based records for a large fraction of the attractions, particularly those that are outdoors. This may provide a powerful new tool for forecasting tourism demands, helping tourism accommodations to get prepared even when there is no prior survey for their regions (or one is not even possible), only by using freely available social media data empowered by environmental records. However, correlating environmental and social data, including recent and seasonal, with official visits demonstrated to be key to motivate the simplicity of our prediction model. One needs to perform such analysis in a much higher scale in the future to determine the real practical and economical benefits of the proposed techniques.

## 5 Conclusions

We have investigated the impact of exploiting recency and seasonality features alongside social media and environmental data to improve the performance of specialized prediction models for touristic attractions (indoor and outdoor). Our experiments showed that by using specialized SVR models including all the tourism requirements, specially the explicit use of recency and seasonality features—outperforms all the baselines, including state-of-the-art solutions [5]. Improvements were obtained in all scenarios, mainly for highly accurate predictions (MAPE $< 10\%$) with gains of more than 300% over the previous solutions.

We have also analyzed the impact of each of the tourism prediction requirements individually and their interactions applying a $2^k$ factorial design analysis. We quantified the performance of each of the three tourism prediction factors (requirements) in the learned models, observing the higher impact of model specialization and seasonality features in model accuracy. But even the less impacting recency features can increase the accuracy of the models, mainly when there is no available seasonal data for a given attraction.

Furthermore, to have a deeper understanding of the impact of the recency and seasonality aspects, we analyzed how scarcity in historical recent and seasonal data impacts the prediction accuracy of models. The general observation was that recent trends of visitation are essential in the accuracy of the models. Finally, we showed how explicit incorporation of seasonality and recency features into the specialized models of indoor and outdoor attractions could improve the accuracy of the tourism demand in attractions in which the state-of-the-art models could not provide an accurate prediction.

### 5.1 Future work

In future work, we intend to continue improving accuracy, mainly of highly accurate predictions (MAPE < 10%), by evaluating the contents and sentiments of the reviews of each attraction. We intend to apply text analysis techniques such as Temporal Topic Modeling and Sentiment Analysis in order to extract useful information from visitors daily reviews and their possible visiting behaviour trends. Another possible research direction is to cluster attractions into a few groups in order to create specific prediction models for each cluster, making it simpler and more practical to use our solutions in the real life of business owners. This could bring a lot of benefits specially by producing robust forecasting models for touristic places with low availability of visitation census, due to multiple reasons such as high costs of surveys or difficulty to collect data in remote places.

## Appendices

This section includes the results of Global model application in the first part while the second part presents the statistical analysis of our factorial design experiments.

### A. Global model (Model specialization = OFF)

The application of model specialization may be considerably jeopardized when there is not enough data to train individual models for each site. In this case, it is more viable to train and apply a single global model taking advantage of the complete social and environmental (training) data for multiple attractions. In here, we evaluate the prediction power of trained global models augmented with seasonality and recency tourism features for each type of tourism attraction—indoors and outdoors. Table 14 and 15 shows these results. In the case of indoor attractions, the global model with only social and environmental features (global mdl.) has a good MAPE (MAPE < 25%) only for about 11% of museums with a similar scenario for outdoor attractions, where there is only about 18% of parks with good prediction results using a global model.

**Table 14. Prediction results adopting seasonality and/or recency tourism requirements for a global model trained with 27 museums in U.K. (indoors).** The best prediction models are in bold face.

| MAPE | Museums—Global Model | | | |
| --- | --- | --- | --- | --- |
| | global mdl. | global mdl.+recency | global mdl.+seasonality | global mdl.+recency+seasonality |
| MAPE<10 | 3.7% | 7.41% | **25.93%** | 7.41% |
| MAPE<25 | 11.11% | 48.15% | **77.78%** | 74.07% |

**Table 15. Prediction results adopting seasonality and/or recency tourism requirements for a global model trained with 76 national parks in U.S. (outdoors).** The best prediction models are in bold face.

| MAPE | Parks—Global Model | | | |
|---|---|---|---|---|
| | global mdl. | global mdl.+recency | global mdl.+seasonality | global mdl.+recency+seasonality |
| MAPE<10 | 5.26% | 3.95% | **30.26%** | **30.26%** |
| MAPE<25 | 18.42% | 46.05% | 81.58% | **86.84%** |

It can also be observed in the Tables that introducing recency and seasonality as features into the global models significantly improves the accuracy of the prediction task. Global models produce good predictions (MAPE < 25) for about 74% of the museums and 87% of the parks. Those results however, are worse than when specialization is applied (if data availability allows), mainly for highly accurate predictions (MAPE < 10). In any case, the good accuracy provided by the global models with recency and seasonality encourage its application for the cases in which there is a lack of enough training data for specific attractions.

**B. Statistical analyses.** The statistical significance of our results is presented (according to a t-test with significance level of $\alpha = 0.05$). Specifically, we report 95% confidence intervals for the effect of each tourism requirement factor on the prediction task (according to our factorial analysis presented in Section 3.3) for both considered scenarios of indoor and outdoor attractions. These are shown in Tables 16 and 17, respectively. In general terms, we find that, with the specified statistical significance, model specialization and seasonality have the largest contributions.

**Table 16. Contribution of each of tourism prediction requirements: Recency, seasonality, model specialization and their interactions into the response variable in indoor attractions (U.K. Museums and Galleries); results for MAPE < 10 and MAPE < 25 in 5 runs.** Minimum and maximum confidence interval for 95% confidence are reported.

| Requirements/Factors | MAPE < 25 | | | MAPE < 10 | | |
|---|---|---|---|---|---|---|
| | contri (%) | CI min | CI max | contri (%) | CI min | CI max |
| A = recency | 1.0 | 0.1 | 3.6 | 0.0 | 0.0 | 0.3 |
| B = seasonality | 21.1 | 12.4 | 38.1 | 19.8 | 12.5 | 31.3 |
| C = model spec. | 55.9 | 36.8 | 92.2 | 70.0 | 51.3 | 98.2 |
| AB | 5.5 | 4.4 | 6.8 | 1.8 | 0.6 | 3.3 |
| AC | 0.1 | 0.1 | 0.6 | 1.3 | 0.3 | 3.9 |
| BC | 13.3 | 13.2 | 14.0 | 2.9 | 1.0 | 6.5 |
| ABC | 0.7 | 0.0 | 2.9 | 1.6 | 0.3 | 4.2 |
| error | 2 | - | - | 3 | - | - |

**Table 17. Contribution of each of tourism prediction requirements: Recency, seasonality, model specialization and their interactions into the response variable in outdoor attractions (U.S. National Parks); results for MAPE < 10 and MAPE < 25 in 5 runs.** Minimum and maximum confidence interval for 95% confidence are reported.

| Requirements/Factors | MAPE < 25 | | | MAPE < 10 | | |
|---|---|---|---|---|---|---|
| | contri (%) | CI min | CI max | contri (%) | CI min | CI max |
| A = recency | 1.1 | 0.7 | 1.5 | 1.1 | 0.5 | 2.2 |
| B = seasonality | 24.6 | 21.5 | 28.3 | 32.6 | 26.0 | 41.2 |
| C = model spec. | 46.0 | 40.9 | 52.0 | 58.5 | 48.1 | 71.9 |
| AB | 0.7 | 0.5 | 0.9 | 2.0 | 2.8 | 1.4 |
| AC | 1.2 | 1.0 | 1.0 | 2.0 | 1.1 | 3.5 |
| BC | 24.4 | 24.0 | 24.8 | 1.8 | 1.2 | 2.6 |
| ABC | 1.8 | 1.4 | 2.5 | 0.9 | 0.5 | 1.5 |
| error | 0.2 | - | - | 1.0 | - | - |

## Author Contributions

**Conceptualization:** Amir Khatibi.

**Data curation:** Amir Khatibi.

**Formal analysis:** Amir Khatibi.

**Funding acquisition:** Amir Khatibi.

**Investigation:** Amir Khatibi.

**Methodology:** Amir Khatibi.

**Project administration:** Amir Khatibi.

**Resources:** Amir Khatibi.

**Software:** Amir Khatibi.

**Supervision:** Amir Khatibi, Ana Paula Couto da Silva, Jussara M. Almeida, Marcos A. Gonçalves.

**Validation:** Amir Khatibi.

**Visualization:** Amir Khatibi.

**Writing – original draft:** Amir Khatibi.

**Writing – review & editing:** Amir Khatibi, Ana Paula Couto da Silva, Jussara M. Almeida, Marcos A. Gonçalves.

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
