## [Decision Letter · Decision Letter 0]

2 May 2022

PONE-D-22-04884A Quantitative Analysis of the Impact of Explicit Incorporation of Recency, Seasonality and Model Specialization into Fine-Grained Tourism Demand Prediction ModelsPLOS ONE

Dear Dr. Khatibi,

Thank you for submitting your manuscript to PLOS ONE. After careful consideration, we feel that it has merit but does not fully meet PLOS ONE’s publication criteria as it currently stands. Therefore, we invite you to submit a revised version of the manuscript that addresses the points raised during the review process.

We look forward to receiving your revised manuscript.

Kind regards,

Ali Safaa Sadiq

Academic Editor

PLOS ONE

Journal Requirements:

2. In your Methods section, please include additional information about your dataset and ensure that you have included a statement specifying whether the collection method complied with the terms and conditions for the website." 2) "Please note that PLOS ONE has specific guidelines on code sharing for submissions in which author-generated code underpins the findings in the manuscript. In these cases, all author-generated code must be made available without restrictions upon publication of the work. Please review our guidelines at https://journals.plos.org/plosone/s/materials-and-software-sharing#loc-sharing-code and ensure that your code is shared in a way that follows best practice and facilitates reproducibility and reuse.

(Applying to all authors, this work is partially supported by CNPq, CAPES, and Fapemig)

(This work is partially supported by CNPq, CAPES, and Fapemig)

(Applying to all authors, this work is partially supported by CNPq, CAPES, and Fapemig)

7. Please update your submission to use the PLOS LaTeX template. The template and more information on our requirements for LaTeX submissions can be found at http://journals.plos.org/plosone/s/latex.

8. We note you have included a table to which you do not refer in the text of your manuscript. Please ensure that you refer to Table 12 and 13 in your text; if accepted, production will need this reference to link the reader to the Table.

Reviewers' comments:

Reviewer's Responses to Questions

**Comments to the Author**

1. Is the manuscript technically sound, and do the data support the conclusions?

Reviewer #1: Yes

Reviewer #2: Yes

Reviewer #3: Yes

2. Has the statistical analysis been performed appropriately and rigorously? 

Reviewer #1: Yes

Reviewer #2: Yes

Reviewer #3: Yes

3. Have the authors made all data underlying the findings in their manuscript fully available?

Reviewer #1: Yes

Reviewer #2: Yes

Reviewer #3: Yes

4. Is the manuscript presented in an intelligible fashion and written in standard English?

Reviewer #1: Yes

Reviewer #2: Yes

Reviewer #3: Yes

5. Review Comments to the Author

Reviewer #1: The contribution of this paper is good and I am happy to endorse its acceptance at some point. However, there are several major and minor comments to address. I have listed them as follows:

• First off, please clearly state the gap targeted in this paper at the end of introduction and list down the hypotheses

• In terms of research method and design, there is not much in the paper.

• The comparative algorithms in the experiments are not properly acknowledged and cited

• I also suggest adding some figures to better articular the content as the paper looks very dry at the moment.

• Analysis of the results is missing in the paper. There is a big gap between the results and conclusion. There should be the result analysis between these two sections. After comparing the numerical methods, you have to be able to analyse the results and relate them to their structures. It would be interesting to have your thoughts on why the method works that way? Such analyses would be the core of your work where you prove your understanding of the reason behind the results. You can also link the findings to the hypotheses of the paper. Long story short, this paper requires a very deep analysis from different perspectives

• There is no statistical test to judge about the significance of the numerical method’s results. Without such a statistical test, the conclusion cannot be supported

• There is no discussion on the cost effectiveness of the proposed method. What is the computational complexity? What is the runtime? Please include such discussions. You can also use the big oh notation to show the computation complexity.

• Some mathematical notations and Lemma presentations are not rigorous enough to correctly understand the contents of the paper. The authors are requested to recheck all the definition of variables and further clarify these equations.

Reviewer #2: This work is a novel work on tourist prediction with the pre-requirements. But several issues should be critical improved for resubmission. I encourage authors to make thoroughful revision and re-submission.

1. This topic is interesting in Tourism management or visiting prediction. However, the research question or focused scientific / technical question didn’t clarity.

2. Structure of manuscript is vague. For example, works on investigating were in Section 1 and Section 2. Some discussion content was included in Section Result.

3. For the references, it is a narrow review and insufficient. The newest reference is published in 2019. From many journals, like p-one, IEEE Access, you can get many works about toursim with keyword="Fine-Grained" or "tourist" or "toursim".

4.The proposed methods with three key requirements for prediction are helpful to improve grained prediction. However, the suitability and limitations of this work should be tested and written.

Reviewer #3: The abstract is too short and not structured according to the standard format.

The manuscript lack literature review on the topic under discussion. I recommend to add a literature review section to highlight the past related work and how your work is different.

Conclusions and Future Work should be separated into two sections.

Add more up-to-date references.

6. PLOS authors have the option to publish the peer review history of their article (what does this mean?). If published, this will include your full peer review and any attached files.

Reviewer #1: No

Reviewer #2: No

Reviewer #3: No

---

## [Author Response · Author response to Decision Letter 0]

23 Jul 2022

We are also sending a pdf file of response letter in complete with our submission;

################################################################

Paper title: A Quantitative Analysis of the Impact of Explicit Incorporation of Recency, Seasonality and Model Specialization into Fine-Grained Tourism Demand Prediction Models

Autores: Amir Khatibi, Ana Paula Couto da Silva, Jussara Almeida, Marcos A. Gonçalves

Dear Editor in Chief and Reviewers

We are sending a revised version of our paper under submission to PLOS ONE Journal as well as the responses to the reviewers’ comments. 

We would like to sincerely thank all reviewers for their thoughtful and detailed comments that helped improve our paper. We did our best to address all their comments. The original reviewers' comments are in bold, each followed by our responses in italic. 

We also performed all the requested editorial changes in our revised manuscript. 

Best regards,

Amir Khatibi,

Ana Paula Couto da Silva, Jussara Almeida, Marcos A. Gonçalves

################################################################

REVIEWER #1

COMMENT #1: 

The contribution of this paper is good and I am happy to endorse its acceptance at some point. However, there are several major and minor comments to address. I have listed them as follows:

First off, please clearly state the gap targeted in this paper at the end of introduction and list down the hypotheses 

RESPONSE: We thank the reviewer for the positive feedback on our paper. 

As requested, we better elaborated the hypotheses that we investigate in this work as explicit research questions we aim to answer throughout the manuscript.

The questions are explicitly stated in a new subsection (Subsection 1.1, pages 4 and 5) alongside the main contributions (Section 1.3, pages 9 and 10) at the end of the Introduction. 

In more detail, in Research Question 1 (RQ1), we investigate whether recency, seasonality and model specialization (characteristic of attraction) do indeed influence the accuracy of predicting visits in tourist sites. In RQ2, we aim to quantify the impact of each of these factors on tourism demand prediction. Finally in RQ3, we study the extent to which scenarios with data scarcity hinder the accuracy of prediction models while exploiting recency, seasonality and model specialization.

The combined answers to these questions advance the state-of-the-art in the field by providing not only a better understanding of the role of the aforementioned factors in the prediction models but also by allowing us to produce prediction models that are more accurate than the current state-of-the-art.

COMMENT #2: 

In terms of research method and design, there is not much in the paper.

RESPONSE: We better defined our methodology and experimental design in the revised version by including new Figures and expanded explanations. For instance, in Figure 2 (page 15) we discuss our tourism demand prediction methodology that adopts social media and environmental data as well as embedded recency and seasonality features in both specialized and global models. In Figures 4 and 5 (pages 24 and 25 respectively), we illustrate how we refine our prediction methodology for two scenarios of scarcity in recent and historical data. We also added a new paragraph in Section 2.5 (page 15) alongside Figure 3 (in the same page) to elaborate the differences between specialized and global models in our methodology.

We hope with these new clarifications, it becomes clearer that we have a well defined, solid and statistically sound methodology and experimental design, which also include experiments with folded cross-validation and statistical tests presented in Appendix (Tables 16 and 17), to better assess the generalization of the learned model under different training and test sets. We also used a factorial design analysis as explained in Section 2.6, to quantify the impact of the analyzed factors.

COMMENT #3: 

The comparative algorithms in the experiments are not properly acknowledged and cited

RESPONSE: We thank the reviewer for the comment. We have revised the manuscript to properly acknowledge and cite all algorithms. 

Notably, in our Prediction Techniques Section (Section 2.3), we presented five prediction algorithms (Support Vector Regression [Drucker 1996], Seasonal ARIMA [Hilmar 1991], Seasonal ARIMAX [Vagro. 2016], Neural Network [Hoch. 1997], Naive Models [McLau. 1986]). As the reviewer suggested, we added a new paragraph properly acknowledging all models with a detailed explanation of calculations and we also revised the SARIMAX model adding necessary references as well. 

Moreover, we also added a paragraph in the Factorial Design analysis section (Section 2.6) citing one of the most widely used books describing its use [Jain 1991].

[Drucker 1996]: Drucker H, Burges CJC, Kaufman L, Smola A, Vapnik V. Support Vector Regression Machines. In: NIPS; 1996. p. 155–161

[Hilmar 1991]: Hillmer SC. Time Series Analysis Univariate and Multivariate Methods. Journal of the American Statistical Association. 1991; p. 245.

[Vagro. 2016]: Vagropoulos ea Stylianos I. Comparison of SARIMAX, SARIMA, modified SARIMA and ANN-based models. IEEE International Energy Conference (ENERGYCON). 2016

[Hoch. 1997]: Hochreiter S, Schmidhuber J. Long Short-term Memory. Neural computation. 1997; vol 9:1735–80.

[McLau. 1986]: McLaughlin, Robert L. Forecasting models: Sophisticated or naive? Journal of Forecasting (pre-1986) 2.3 (1983): 274.

[Jain 1991]: Jain Rea. A test of goodness of fit. The Art of Computer Systems Performance Analysis: techniques for experimental design, measurement, simulation, and modeling. 1991;49(268):765–769

COMMENT #4: 

I also suggest adding some figures to better articular the content as the paper looks very dry at the moment.

 RESPONSE: We thank the reviewer for the suggestion. 

We have added new figures to make it easy for the reader to follow our manuscript. Specifically, we added:

a new Figure 1 in Section 2 (page 11) to better illustrate the phases of our data collection methodology, and

a new Figure 3 in Section 2 (page 15) to better illustrate the differences between the specialized and global models in an illustrative demonstration.

We have also improved the figure that described the temporal evolution of tourism features (previously Figure 4) breaking it into two new more informative Figures (now Figures 6 and 7).

We believe that these new figures, along with the figures already available in the original version of the manuscript describing various methodological aspects of our study will make the reading much easier. 

COMMENT #5: 

Analysis of the results is missing in the paper. There is a big gap between the results and conclusion. There should be the result analysis between these two sections. After comparing the numerical methods, you have to be able to analyze the results and relate them to their structures. It would be interesting to have your thoughts on why the method works that way? 

RESPONSE: As suggested, we added a new Section (Section 4 - Discussions) following the Results sections, to discuss and analyze our achieved results in a more qualitative fashion. 

In the new section, we better relate the obtained experimental results, described in Section 3, with the posed research questions in Section 1, connecting the hypotheses with experimental results more clearly. Finally, we offered insights into why the method works the way it did based on our interpretations of all performed analyses.

COMMENT #6: 

Such analyses would be the core of your work where you prove your understanding of the reason behind the results. You can also link the findings to the hypotheses of the paper. Long story short, this paper requires a very deep analysis from different perspectives

RESPONSE: As we mentioned in our response to the previous comment, we have included a new Section (Section 4 - Discussions) where we perform a deeper analysis of our findings, better associating our research questions (a translation of our hypotheses) with the experimental results reported in the paper.

COMMENT #7: 

There is no statistical test to judge about the significance of the numerical method’s results. Without such a statistical test, the conclusion cannot be supported

 RESPONSE: We have added a new section - Statistical analysis in the Appendix - on the statistical significance of our results (according to a t-test with significance level alpha = 0.05). Specifically, we report 95% confidence intervals for the effect of each tourism requirement factor on the prediction task (according to our factorial analysis presented in Section 3.3) for both considered scenarios of indoor and outdoor attractions. These are shown in Tables 16 and 17, respectively. In general terms, we find that, with the specified statistical significance, model specialization and seasonality have the largest contributions.

COMMENT #8: 

There is no discussion on the cost effectiveness of the proposed method. What is the computational complexity? What is the runtime? Please include such discussions. You can also use the big oh notation to show the computation complexity.

 RESPONSE: We added a new paragraph in Section 3.1 (pages 18 and 19), reporting computational complexity of the main prediction algorithm and execution times for specialized and global models. 

The Support Vector Machine (SVR) has a time complexity of O(n3) and space complexity of O(n2) where N is the number of points [Hui 2016]. In our experiments, since we have few training points (n = 30 for outdoor attractions and n = 76 in indoor attractions), and our model is trained once a month, the execution time for the prediction task was not a major concern. 

The computational complexity of the other algorithms is as follows. For SARIMA models the complexity is in the order of O(n). Neural network models, in turn, take more time due to numerous iterations applying forward and back-propagation — Backpropagation, in the order of O(n5), is much slower than the forward propagation in the order of O(n4). Finally, for the case of naive models the complexity is O(1) since they only pick the defined index of historical data to pass as the naive prediction.

Regarding execution time, for specialized SVR models, the mean execution time is about 45 seconds (min exec. time: 21 seconds and max exec. time: 92 seconds) while for generalized models for different sets of features the mean execution time was around 4 hours (min exec time: 1h and 33 minutes and max exec time: 20 hours). After the conclusion of the training step, the prediction phase is quite fast (average of 4 seconds for all 100 attractions independent of global or specialized models).

Observation: As we also reported in the revised version of the paper, for our experiments, we used a desktop PC with 4 CPUs and 16 GBs of RAM memory using the R programming language.

[Hui 2016]: Y. Hui, S. Wenzhu, Z. Xiuzhi, Z. Guotao and H. Wenting, Heuristic sample reduction based support vector regression method, IEEE International Conference on Mechatronics and Automation, 2016, pp. 2065-2069,

COMMENT #9: 

Some mathematical notations and Lemma presentations are not rigorous enough to correctly understand the contents of the paper. The authors are requested to recheck all the definition of variables and further clarify these equations.

RESPONSE: We revisited all the mathematical definitions in Sections 2.2 Problem Definition and 2.3 Prediction Techniques in order to make sure that all notations are correct and consistent. There are no lemmas in our manuscript.

################################################################

REVIEWER #2

COMMENT #1: 

This work is a novel work on tourist prediction with the pre-requirements. But several issues should be critical improved for resubmission. I encourage authors to make thoroughful revision and re-submission.

This topic is interesting in Tourism management or visiting prediction. However, the research question or focused scientific / technical question didn’t clarity.

RESPONSE: We thank the reviewer for the comment. 

As stated in our responses to Comments #1 and #6 of Reviewer #1, to clarify our research questions, we added a new Section at the end of the introduction of the revised manuscript where we explicitly state and explain our Research Questions – RQs (new Section 1.1, pages 4 and 5). We then connect these RQs to the experimental results in the paper with a qualitative analysis in Section 4 - Discussions.

COMMENT #2: 

Structure of manuscript is vague. For example, works on investigating were in Section 1 and Section 2. Some discussion content was included in Section Result.

RESPONSE: We followed the PLOS One required structure.

According to the PLOS ONE submission guideline (link), we added our related work study in Introduction - Section 1:

PLOS ONE: 

 - The introduction should: Include a brief review of the key literature.

- The Materials and Methods section: If materials, methods, and protocols are well established, authors may cite articles where those protocols are described in detail, but the submission should include sufficient information to be understood independent of these references.

Regarding the inclusion of discussion content in Section Results, as indicated in our response to Comment #5 of Reviewer #1, we added a new section, Section 4 - Discussions where we concentrated (and expanded) all the discussions and qualitative analysis of the results. 

We also changed the title of Section ‘Experimental Methodology’ to ‘Materials and Methods’ to closely comply with the PLOS ONE guidelines.

COMMENT #3: 

For the references, it is a narrow review and insufficient. The newest reference is published in 2019. From many journals, like p-one, IEEE Access, you can get many works about toursim with keyword="Fine-Grained" or "tourist" or "toursim".

RESPONSE: As the reviewer suggested, we searched for recent works in the area of tourism demand prediction. We included the related papers in well-known journals such as IEEE Access and PLOS ONE published in recent years. The citations to three recently published works were included in the Related Work (Section 1.2), describing each work and their similarities and differences to ours. 

In more details, In [Luo 2021], authors investigate experiences of Chinese economy hotel guests using online reviews as proxy; similarly to our work, the authors use external data for fine-grained predictions. However, the focus of our work is in visits prediction instead of sentiment analysis. We also exploit recent and seasonal behaviors explicitly in our feature-set. 

[Li 2021] develops a scalable online platform for extracting, analyzing, and sharing multi-source multi-scale human mobility flows to assist human mobility monitoring. The focus of this work is mostly on providing and monitoring fine-grained spatio-temporal mobility data while in our work we analyze multiple prediction models exploiting external data alongside explicit use of seasonality and recency in order to predict tourism demand.

In [Zhang 2021], authors build a fine-grained tourist satisfaction prediction model based on deep learning, using destination features as proxy; similarly to our work, the authors use external data for fine-grained prediction. However the focus of our work is in visits prediction instead of tourist satisfaction prediction. In addition, in our work, we exploit recent and seasonal features in order to obtain more accurate results.

[Luo 2021]: Luo J, Huang SS, Wang R. A fine-grained sentiment analysis of online guest reviews of economy hotels in China. Journal of Hospitality Marketing & Management. 2021;30(1):71–95

[Li 2021]: Li ea Zhenlong. ODT FLOW: Extracting, analyzing, and sharing multi-source multi-scale human mobility. Journal of Plos one. 2021;16(8)

[Zhang 2021]: Zhang H, Wang Z, Ke M, Cai M, Sun Q. Fine-grained Tourist Satisfaction Prediction Based on Deep Learning, 3rd International Conference on Frontiers Technology of Information and Computer (ICFTIC); IEEE 2021. p. 30–35.

COMMENT #4: 

The proposed methods with three key requirements for prediction are helpful to improve grained prediction. However, the suitability and limitations of this work should be tested and written.

RESPONSE: We added a new Limitations subsection (Section 4.1) in the Discussion Section specifying the limitations of our analysis such as lack of official data in order to test our methodology on finer time-grained data (weekly or daily basis). Another limitation is the scale of our study that, though larger than most similar studies, still, needs to be expanded to assess the practicality of our proposals in the real-world.

################################################################

REVIEWER #3

COMMENT #1: 

 The abstract is too short and not structured according to the standard format.

RESPONSE: According to the PLOS ONE submission guideline (link), an abstract should not exceed 300 words. Our abstract was structured to comply with this guideline. 

COMMENT #2: 

The manuscript lack literature review on the topic under discussion. I recommend to add a literature review section to highlight the past related work and how your work is different.

RESPONSE: We have analyzed more than sixteen (16) works in the literature in the area of tourism prediction where all are classified and presented in Table 1 in the Introduction section. As stated in our response to Comment #2 of Reviewer #2, according to the PLOS ONE guideline (link), the literature review should be included in the Introduction Section. 

Now, in the revised version we have an explicit literature review Section (Section Related Work, 1.2, pages 5, 6, 7, 8 and 9) as a subsection of the Introduction as required by the PLOS One format guidelines. We also expanded the number of works in this Section, including three recent, very related studies.

COMMENT #3: 

Conclusions and Future Work should be separated into two sections.

RESPONSE: According to the PLOS ONE submission guideline (link), we could not have a different Section for Future Work:

PLOS ONE: Results, Discussion, Conclusions These sections may all be separate, or may be combined to create a mixed Results/Discussion section. 

Authors should explain how the results relate to the hypothesis presented as the basis of the study and provide a succinct explanation of the implications of the findings, particularly in relation to previous related studies and potential future directions for research.

We separated Conclusions and Future Work into two subsections in the same Section of Conclusion, following the PLOS ONE submission guideline. 

COMMENT #4: 

Add more up-to-date references.

RESPONSE: As stated in our response to Comment #3 of Reviewer #2, after searching for recent tourism demand prediction publications in well-known journals, we added three more citations to Table 1 in our Related Work (Section 1.2, pages 5, 6, 7, 8 and 9). 

Luo J, Huang SS, Wang R. A fine-grained sentiment analysis of online guest reviews of economy hotels in China. Journal of Hospitality Marketing & Management. 2021;30(1):71–95

Li ea Zhenlong. ODT FLOW: Extracting, analyzing, and sharing multi-source multi-scale human mobility. Journal of Plos one. 2021;16(8)

Zhang H, Wang Z, Ke M, Cai M, Sun Q. Fine-grained Tourist Satisfaction Prediction Based on Deep Learning, 3rd International Conference on Frontiers Technology of Information and Computer (ICFTIC); IEEE 2021. p. 30–35.

---

## [Decision Letter · Decision Letter 1]

16 Aug 2022

PONE-D-22-04884R1A Quantitative Analysis of the Impact of Explicit Incorporation of Recency, Seasonality and Model Specialization into Fine-Grained Tourism Demand Prediction ModelsPLOS ONE

Dear Dr. Khatibi,

Thank you for submitting your manuscript to PLOS ONE. After careful consideration, we feel that it has merit but does not fully meet PLOS ONE’s publication criteria as it currently stands. Therefore, we invite you to submit a revised version of the manuscript that addresses the points raised during the review process.

We look forward to receiving your revised manuscript.

Kind regards,

Ali Safaa Sadiq

Academic Editor

PLOS ONE

Journal Requirements:

Additional Editor Comments (if provided):

Authors are invited to submit their second revision of their manuscript to address some of the minor changes suggested by the first reviewer.

Reviewers' comments:

Reviewer's Responses to Questions

**Comments to the Author**

1. If the authors have adequately addressed your comments raised in a previous round of review and you feel that this manuscript is now acceptable for publication, you may indicate that here to bypass the “Comments to the Author” section, enter your conflict of interest statement in the “Confidential to Editor” section, and submit your "Accept" recommendation.

Reviewer #1: (No Response)

Reviewer #3: All comments have been addressed

2. Is the manuscript technically sound, and do the data support the conclusions?

Reviewer #1: (No Response)

Reviewer #3: Yes

3. Has the statistical analysis been performed appropriately and rigorously? 

Reviewer #1: (No Response)

Reviewer #3: Yes

4. Have the authors made all data underlying the findings in their manuscript fully available?

Reviewer #1: (No Response)

Reviewer #3: Yes

5. Is the manuscript presented in an intelligible fashion and written in standard English?

Reviewer #1: (No Response)

Reviewer #3: Yes

6. Review Comments to the Author

Reviewer #1: Some final cosmetic comments:

* The results of your comparative study should be discussed in-depth and with more insightful comments on the behaviour of your algorithm on various case studies. Discussing results should not mean reading out the tables and figures once again.

* Avoid lumping references as in [x, y] and all other. Instead summarize the main contribution of each referenced paper in a separate sentence. For scientific and research papers, it is not necessary to give several references that say exactly the same. Anyway, that would be strange, since then what is innovative scientific contribution of referenced papers? For each thesis state only one reference.

* Avoid using first person.

* Avoid using abbreviations and acronyms in title, abstract, headings and highlights.

* Please avoid having heading after heading with nothing in between, either merge your headings or provide a small paragraph in between.

* The first time you use an acronym in the text, please write the full name and the acronym in parenthesis. Do not use acronyms in the title, abstract, chapter headings and highlights.

* The results should be further elaborated to show how they could be used for the real applications.

Reviewer #3: Wish a very good luck for all authors. The manuscript now is ready for publication after addressing all the comments.

7. PLOS authors have the option to publish the peer review history of their article (what does this mean?). If published, this will include your full peer review and any attached files.

Reviewer #1: No

Reviewer #3: No

---

## [Author Response · Author response to Decision Letter 1]

3 Oct 2022

Paper title: A Quantitative Analysis of the Impact of Explicit Incorporation of Recency, Seasonality and Model Specialization into Fine-Grained Tourism Demand Prediction Models

Autores: Amir Khatibi, Ana Paula Couto da Silva, Jussara Almeida, Marcos A. Gonçalves

Dear Plos One Editor in Chief and Reviewers

We are sending a second revision version of our manuscript under submission to PLOS ONE Journal as well as the responses to the reviewers’ comments. 

We would like to sincerely thank all reviewers for their thoughtful and detailed comments that helped improve our paper. We did our best to address all their comments. The original reviewers' comments are in bold, each followed by our responses in italic. 

We also performed all the requested editorial changes in our revised manuscript. 

Best regards,

Amir Khatibi,

Ana Paula Couto da Silva, Jussara Almeida, Marcos A. Gonçalves

REVIEWER #1 

 Some final cosmetic comments:

COMMENT #1: 

The results of your comparative study should be discussed in-depth and with more insightful comments on the behavior of your algorithm on various case studies. Discussing results should not mean reading out the tables and figures once again.

RESPONSE: We thank the reviewer for the feedback on our paper. 

As requested, we expanded the Discussion Section with more insightful comments on our experimental results. We also avoided re-reading previous tables and figures and focused on providing more qualitative comments based on our case studies, as suggested. 

COMMENT #2:

Avoid lumping references as in [x, y] and all others. Instead summarize the main contribution of each referenced paper in a separate sentence. For scientific and research papers, it is not necessary to give several references that say exactly the same. Anyway, that would be strange, since then what is the innovative scientific contribution of referenced papers? For each thesis state only one reference.

RESPONSE: We checked for all lumping references in the Related Work Section and broke them to explain the contribution of each separately.

COMMENT #3:

Avoid using first person.

RESPONSE: We reduced considerably the use of the first person in the revised manuscript, although we did not remove all uses, since in some points of the text it would sound unnatural or artificial. 

COMMENT #4:

Avoid using abbreviations and acronyms in title, abstract, headings and highlights.

RESPONSE: We resolved the cases using acronyms in sub-sections in Section 2.4.

COMMENT #5:

Avoid having heading after heading with nothing in between, either merge your headings or provide a small paragraph in between.

RESPONSE: We reviewed the paper to guarantee avoiding the issues emphasized by the reviewer. In addition, in the Appendices Section, we made some modifications to avoid heading after dividing the Appendices Section into part A and B with a small paragraph explaining each of these sub-sections.

COMMENT #6:

The first time you use an acronym in the text, please write the full name and the acronym in parenthesis. Do not use acronyms in the title, abstract, chapter headings and highlights.

RESPONSE: We revised the paper and took care of three incidents of this type. Now every acronym correctly appears in parentheses after the full name.

COMMENT #7:

The results should be further elaborated to show how they could be used for the real applications.

RESPONSE: We expanded the discussion in Section 4.2, emphasizing limitations of our study and discussing issues related to the practical applications of our results, analyses and findings.

REVIEWER #2

There were no comments from Reviewer #2.

REVIEWER #3

Wish a very good luck for all authors. The manuscript now is ready for publication after addressing all the comments.

RESPONSE: We thank the reviewer #3 for kindful feedback and good wishes.

---

## [Decision Letter · Decision Letter 2]

10 Nov 2022

A Quantitative Analysis of the Impact of Explicit Incorporation of Recency, Seasonality and Model Specialization into Fine-Grained Tourism Demand Prediction Models

PONE-D-22-04884R2

Dear Dr. Khatibi,

We’re pleased to inform you that your manuscript has been judged scientifically suitable for publication and will be formally accepted for publication once it meets all outstanding technical requirements.

Kind regards,

Ali Safaa Sadiq

Academic Editor

PLOS ONE

Additional Editor Comments (optional):

The authors have addressed all the given comments by reviewers, hence I am happy to recommend their paper for the possible publication.

Reviewers' comments:

Reviewer's Responses to Questions

**Comments to the Author**

1. If the authors have adequately addressed your comments raised in a previous round of review and you feel that this manuscript is now acceptable for publication, you may indicate that here to bypass the “Comments to the Author” section, enter your conflict of interest statement in the “Confidential to Editor” section, and submit your "Accept" recommendation.

Reviewer #1: (No Response)

2. Is the manuscript technically sound, and do the data support the conclusions?

Reviewer #1: (No Response)

3. Has the statistical analysis been performed appropriately and rigorously? 

Reviewer #1: (No Response)

4. Have the authors made all data underlying the findings in their manuscript fully available?

Reviewer #1: (No Response)

5. Is the manuscript presented in an intelligible fashion and written in standard English?

Reviewer #1: (No Response)

6. Review Comments to the Author

Reviewer #1: all comments have been addressed. all comments have been addressed. all comments have been addressed. all comments have been addressed.

7. PLOS authors have the option to publish the peer review history of their article (what does this mean?). If published, this will include your full peer review and any attached files.

Reviewer #1: No

---

## [Editor Report · Acceptance letter]

17 Nov 2022

PONE-D-22-04884R2 

A Quantitative Analysis of the Impact of Explicit Incorporation of Recency, Seasonality and Model Specialization into Fine-Grained Tourism Demand Prediction Models 

Dear Dr. Khatibi:

I'm pleased to inform you that your manuscript has been deemed suitable for publication in PLOS ONE. Congratulations! Your manuscript is now with our production department. 

Kind regards, 

on behalf of

Dr. Ali Safaa Sadiq 

Academic Editor

PLOS ONE